# A channel selection methodology for enhancing volcanic $SO_2$ monitoring using FY-3E/HIRAS-II hyperspectral data

Xinyu Li[1], Lin Zhu[2], Hongfu Sun[1], Jun Li[2], Ximing Lv[1], Chengli Qi[2], Huanhuan Yan[2]

[1]College of Geoscience and Surveying Engineering, China University of Mining and Technology-Beijing, Beijing 100083, China

[2]Key Laboratory of Radiometric Calibration and Validation for Environmental Satellites, National Satellite Meteorological Center (National Center for Space Weather), Beijing 100081, China

*Correspondence to*: Lin Zhu (zhulin@cma.gov.cn)

**Abstract.** The Hyperspectral Infrared Atmospheric Sounder Type II (HIRAS-II) aboard the Fengyun 3E (FY-3E) satellite provides valuable data on the vertical distribution of atmospheric states. However, effectively extracting quantitative atmospheric information from the observations is challenging due to the large number of hyperspectral sensor channels, inter-channel correlations, associated observational errors, and susceptibility of the results to influence by trace gases. This study explores the potential of FY-3E/HIRAS-II to atmospheric loadings of $SO_2$ from volcanic eruptions. A methodology for selecting $SO_2$ sensitive channels from the large number of hyperspectral channels recorded by FY-3E/HIRAS-II is presented. The methodology allows for the selection of $SO_2$-sensitive channels that contain similar information on variations in atmospheric temperature and water vapor for minimizing the influence of atmospheric water vapor and temperature to $SO_2$. A sensitivity study shows that the difference in brightness temperature between the experimentally selected $SO_2$ sensitive channels and the background channels efficiency removes interference signals from surface temperature, atmospheric temperature, and water vapor during $SO_2$ detection and inversion. A positive difference between near-surface atmospheric temperature and surface temperature enables the infrared band to capture more $SO_2$ information in the lower and middle layers. The efficiency of FY-3E/HIRAS-II $SO_2$ sensitive channels in quantitatively monitor volcanic $SO_2$ is demonstrated using data from the 29 April 2024 eruption of Mount Ruang in Indonesia. Using FY-3E/HIRAS-II measurements, the spatial distribution and qualitative information of volcanic $SO_2$ are easily observed. The channel selection can significantly enhance the computation efficiency while maintain the accuracy of $SO_2$ detection and retrieval, despite the large volume of data.

## 1 Introduction

Volcanoes pose significant threats to human populations around the world. During eruptions, they release a variety of gases (e.g., $CO_2$ and $SO_2$), liquids (e.g., $H_2O$ and $H_2SO_4$), and solids (e.g., glass, minerals, and salts), with far-reaching environmental and climatic impacts (Patrick et al., 2020). Understanding the vertical distributions of these substances is essential to analyzing their atmospheric reactions (Bauduin et al., 2017).

Sulfur dioxide ($SO_2$) is a magmatic volatile that is critical to volcanic geochemical analysis and hazard assessment due to its low ambient concentration, high abundance in volcanic plumes, and distinct spectral characteristics (Schmidt et al., 2012). The

1991 eruption of Mount Pinatubo and the 2014 eruption of Mount Bárðarbunga are both significant volcanic $SO_2$ eruption events, each producing $SO_2$ plumes exceeding $1 \times 10^{10}$ kg (Shibata & Kinoshita, 2015). The 1991 Pinatubo eruption in particular produced a plume that peaked at 40 km height, resulting in the largest atmospheric aerosol event since the 1883 Krakatoa eruption (Holasek et al., 1996). Similarly, the 1982 eruption of El Chichón released approximately $7.5 \times 10^9$ kg of $SO_2$ into the atmosphere, reaching 31 km in height (Carey & Sigurdsson, 1986). Tropospheric volcanic $SO_2$ and its transformation products affect the environment, human health, air quality, and the Earth's radiation balance (Gíslason et al., 2015). Hence, systematic monitoring of volcanic $SO_2$ emissions is essential.

Satellite radiometry offers significant advantages for this purpose, including long-term continuity and extensive spatial coverage (Krueger et al., 2009). Ultraviolet (UV) band sensors are limited to monitoring $SO_2$ from daytime eruptions due to their reflective nature. In contrast, general infrared (IR) sensors, with their broader channels, may filter out some $SO_2$ spectral information (Watson et al., 2004). Different techniques have been developed which make use of satellite-based broadband IR channels to detect volcanic $SO_2$ plume (Corradini et al., 2021; Corradini et al., 2010; Doutriaux-Boucher & Dubuisson, 2009; Prata & Kerkmann, 2007; Prata et al., 2004; Tournigand et al., 2020). It is found that the strong absorption at 7.3 μm is heavily affected by low level water vapor and thus this channel is usually used to retrieve $SO_2$ that is high (>3 km) in the atmosphere, and hence above most of the water vapor (Taylor et al., 2018). In addition, the retrieval is also very sensitive to uncertainties on surface temperature and emissivity (Corradini et al., 2009). Meanwhile, wide spectral channels are not sensitive enough to instantaneous changes in $SO_2$ composition, which will increase the minimum concentration of $SO_2$ components that can be monitored (Carn et al., 2003). Hyperspectral IR sensors enable observations with finer channel bandwidths that accurately characterize and distinguish each component, thereby reducing interference from other materials (Milstein & Blackwell, 2016). Although hyperspectral IR sensors provide thousands of spectral channels, they cannot all be used simultaneously for near real-time (NIR) operations owing to unmanageable data volumes and high computational burdens (Li & Han, 2017). At the same time, substantial redundancy and correlation mean that not all channels need to be considered. In addition, the low spectral resolution of traditional multispectral sensors makes it difficult for them to distinguish many important targets (Kruse, 2004) and is limited in quantitative calculations (Feng et al., 2006), thus reducing detection and retrieval accuracy.

To improve computational efficiency and detection accuracy, and to achieve rapid and accurate data acquisition require the selection of a set of channels that provide the maximum amount of information for specific applications (Chang et al., 2020). Rabier et al. (2002) proposed the "constant" iteration method for channel selection for the Infrared Atmospheric Sounding Interferometer (IASI) under clear-sky conditions, which maximized the information for applications. Fourrié and Rabier (2004) selected IASI channels for cloud-sensitive regions based on entropy reduction, demonstrating the robustness of the method Gambacorta and Barnet (2013) used a physical approach to select channels based solely on their spectral characteristics, emphasizing spectral purity, avoiding redundancy, vertical sensitivity, low instrument noise, and global optimality. Lipton (2003) developed a method to select atmospheric microwave sounding channels based on the combination of each channel's center frequency, bandwidth, and degrees of freedom for the signal, with both applicability to multiple environmental conditions and providing robust retrieval performance taken into consideration. Noh et al. (2017) employed the channel score

index to individually evaluate channels selected using a one-dimensional variational (1Dvar) assimilation method. They used entropy subtraction for a comparative study of the selected channels, significantly reducing water vapor errors in the upper troposphere. Ventress and Dudhia (2014) proposed a 1Dvar method for selecting IASI channels and compared it with the method currently employed to choose channels for numerical weather prediction; their method reduced the sensitivity of the channel set to unknown spectral correlations while maintaining the same number of degrees of freedom for the signal. As information entropy iterative techniques do not consider the dynamic impacts of measurements throughout time and only account for the reduction in atmospheric state uncertainty from a single measurement, Di et al. (2022) developed an alternative approach to channel selection for the geostationary hyperspectral IR sounder by incorporating an M-index that considers temporal variations in the variance of the Jacobian. The adapted algorithm improved the accuracy of water vapor profile inversion.

The Jacobian function reflects the sensitivity of the radiation measured at a given pressure level in the atmosphere to changes in substance concentration (Di et al., 2016). In this paper, we propose a channel selection method based on the Jacobian matrix for SO₂ detection and retrieval using the Infrared Hyperspectral Atmospheric Vertical Sounder Type II (HIRAS-II) instrument onboard the Fengyun 3E (FY-3E) satellite.

The remainder of this paper is organized as follows. Section 2 details the data, the radiative transfer principle, and the radiative transfer model employed. Section 3 outlines the methodology of utilizing the Jacobian matrix to select sensitive and background channels for SO₂ monitoring. Section 4 investigates the effects of surface temperature and near-surface air atmospheric temperature variations on SO₂, as well as the sensitivity of detecting SO₂ plumes in the preferred channels. Section 5 demonstrated a case study of Mount Ruang on the comparison of the effectiveness of SO₂ detection between the preferred channels and other absorption channels. Finally, section 6 provides a summary and discussion of the main findings.

## 2 Model and data

### 2.1 Radiative transfer model

The radiation observed by instruments at the top of the atmosphere (TOA) is modulated by the physical properties of both the atmosphere and Earth's surface (Aires et al., 2002). The atmospheric radiative transfer equation is a fundamental framework that governs the behavior of solar electromagnetic radiation and thermal radiation from both the atmosphere and the surface. It is crucial to analyzing radiative transfer processes and understanding atmospheric physical parameters (Seidel et al., 2010). In the absence of scattering and assuming local thermal equilibrium, the atmospheric radiative transfer equation in the IR band can be formulated as follows:

$$R = \varepsilon B_s(T_s)\tau_s - \int_0^{P_s} B(T)d\tau + (1-\varepsilon)\int_0^{P_s} B(T)d\tau^* + 2.16 \times 10^{-5}\pi \cos\theta \times \rho_r B_r(T_{sun}) \times \tau_s^2, \tag{1}$$

where R represents spectral radiation, B is the Planck function at pressure level P, $\tau$ is total atmospheric transmittance above pressure level P, $\varepsilon$ is surface emissivity, $T_s$ is surface temperature, T is the true atmospheric temperature, $\theta$ is the zenith angle,

$\rho_r$ is solar reflectivity, $T_{sun}$ is solar temperature, and define $\tau^* = \tau_s^2/\tau$, (Li, 1994). Among them, subscript s represents surface skin and subscript r represents solar radiation. The term R represents the radiation reaching the satellite. The right-hand side of the equation has four components. The first is the surface emission term, which describes the radiation emitted from the surface that is transmitted through the atmosphere to the satellite. The second term accounts for the upward atmospheric radiation. The third captures the contribution of downward atmospheric radiation reflected from the surface to the satellite. The fourth term represents the contribution of solar radiation to the IR band, which can be neglected here because our focus is on the mid-wave and long-wave IR regions.

To calculate the TOA radiation using Eq. (1), the atmosphere is typically discretized into multiple layers, whose average properties (e.g., temperature, pressure, and molecular species) can be determined. Radiative transfer models facilitate this by allowing precise computation of radiation transmitted through atmospheric gases.

This study uses the Line-By-Line Radiative Transfer Model (LBLRTM), which is a sophisticated, vectorized model derived from Fast Atmospheric Signature Code. LBLRTM can accurately compute atmospheric fluxes and heating rates, making it well-suited to retrieving atmospheric temperature profiles and trace gas concentrations from high-resolution spectral radiance data (Clough, 1994). LBLRTM allows for the input of user-defined atmospheric profile files. In this study, the meteorological data input into LBLRTM consists of six standard atmospheric profiles: the 1976 US Standard Atmosphere, as well as profiles for mid-latitude summer, mid-latitude winter, subarctic summer and subarctic winter (Krueger & Minzner, 1976). These profiles provide 99 vertical levels of atmospheric parameters such as temperature, water vapor concentration, and $SO_2$. Additional inputs include surface temperature, satellite zenith angle, and specific spectral band information, which are essential to calculating the simulated radiance and the Jacobian matrix. Given the spectral absorption characteristics of water vapor, temperature, and $SO_2$ in the IR region, this study focuses on the mid- and long-wave IR bands observed by FY-3E/HIRAS-II.

## 2.2 FY-3E/HIRAS-II data

The FY-3E meteorological satellite is the world's first civilian dawn–dusk orbiting meteorological satellite (Zhang et al., 2022). It is part of China's second-generation polar-orbiting meteorological satellite series. Launched in July 2021, it delivers global cross-spectral atmospheric temperature and humidity vertical distribution data twice daily, in the morning and evening. Working at an inclination of 98.75° and altitude of 836 km, FY-3E completes 14 orbits around the Earth's poles each day, with each orbit taking ~101.5 min, thus achieving comprehensive global coverage after 14 orbits. The satellite's HIRAS-II sensor features 3053 IR channels: 834 long-wave, 1207 mid-wave, and 1012 short-wave. Its measurements span a continuous spectrum range of 648.75 to 2551.25 cm$^{-1}$ at a resolution of 0.625 cm$^{-1}$. Each infrared band contains $3 \times 3$ detector arrays, which simultaneously observe the target area. A complete scanning cycle of HIRAS-II lasts 8 s, the instantaneous field of view (FOV) of each detector to the ground is 1.1°, Fig.1 is a schematic diagram of the field of view (Li et al., 2023). Based on the radiometric specifications for FY-3E/HIRAS-II, the noise equivalent target brightness temperature (BT) difference (NEdT) is specified within 0.2 – 0.4 K for the long-wave IR band, 0.2 – 0.3 K (at 280 K) for the mid-wave IR band and 0.8 – 2.4 K (at 280 K) for the short-wave IR band (Huang et al., 2023). Overall, it delivers high-resolution IR

spectra of the ground–atmosphere system. FY-3E/HIRAS-II data are freely available from the FENGYUN Satellite Data Service (https://satellite.nsmc.org.cn/DataPortal/cn/home/index.html). Table 1 Spectral parameters of FY-3E/HIRAS-II channels (Xie et al., 2023)

| IR Wave Band | Spectral Range (cm$^{-1}$) | No. of Channels | Spectral Resolution (cm$^{-1}$) |
|---|---|---|---|
| Long | 648.75 – 1169.375 (15.41 – 8.55 μm) | 834 | 0.625 |
| Mid | 1167.5 – 1921.25 (8.56 – 5.20 μm) | 1207 | 0.625 |
| Short | 1919.375 – 2551.25 (5.21 – 3.92 μm) | 1012 | 0.625 |

In practical applications, the Level 1 (L1) observation data from HIRAS-II require apodization to mitigate sidelobe effects (Xie et al., 2023). This is accomplished in the present study using the Hamming window function. In addition, radiometric measurements are typically integrated over a wavenumber interval and modified by the instrument's line shape (Crevoisier et al., 2003). Consequently, we convolve the simulated brightness temperature (BT) with the FY-3E/HIRAS-II spectral response function to facilitate subsequent channel selection.

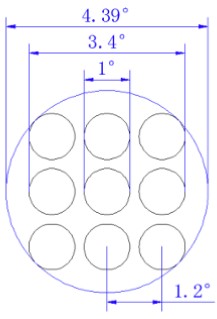

**Figure 1: HIRAS-II detector distribution and corresponding ground field of view.**

### 2.3 Sentinel-5P/TROPOMI SO₂ data

Sentinel-5P is a quasi-polar, sun-synchronous satellite in a low Earth orbit with a height of about 824 km, and it covers the entire planet each day (van Geffen et al., 2020). Every orbital period lasts 16 days, with an average of 227 orbits every period (14 orbits per day) (Corradino et al., 2024). The satellite hosts the Tropospheric Monitoring Instrument (TROPOMI). Daily or sub-daily revisits of specific sites are achievable, given TROPOMI's 108° cross-orbit field of view and its ability to capture data across multiple orbits (Theys et al., 2017). Since 2019, Sentinel-5P's spatial resolution has been enhanced to 3.5 km × 5.5 km. TROPOMI measures data across four spectral regions (ultraviolet, visible, near-infrared, and shortwave

infrared) and is adept at monitoring SO₂ and a range of other gases (Theys et al., 2019). With a comparable footprint of 12 km diameter, TROPOMI demonstrates greater sensitivity to SO₂ variations than IASI (Cofano et al., 2021).

This study uses TROPOMI's Level 2 (L2) geophysical SO₂ products, accessible through the European Space Agency's Copernicus Open Access Center via the Sentinel-5P Pre-Operations Hub. We are using the offline (OFFL) data of this version, which are freely available (Copernicus Sentinel-5P, 2020). These L2 products are derived from Level 0 (L0) raw data, which undergo calibration and georeferencing, followed by processing to Level 1b (L1b) data, including brightness and irradiance. In this study, Sentinel-5P/TROPOMI SO₂ data are primarily employed to validate the SO₂ detection capabilities of FY-3E/HIRAS-II at Mount Ruang (Inness et al., 2022).

## 2.4 Atmospheric profile data

This study employs standard atmospheric profile data as inputs for the LBLRTM. The profiles used are the US Standard Atmosphere, 1976, and tropical, mid-latitude summer and winter, subarctic summer and winter profiles. The US Standard Atmosphere, 1976, serves as an idealized stable representation of Earth's atmosphere from the surface to 1000 km, detailing the relative changes in atmospheric composition with altitude. Below 86 km, the atmospheric composition is calculated using a series of linear functions, while the upper region is defined by continuous functions that closely approximate observational data (Krueger & Minzner, 1976).

ERA5 is the latest comprehensive reanalysis dataset from the European Centre for Medium-Range Weather Forecasts (ECMWF), superseding ERA-Interim. With daily updates, ERA5 provides hourly estimates of the world's atmosphere, land surface, and waves in the ocean from 1950 onward (Hersbach et al., 2020). Each profile from ERA5 has a horizontal scale of 31 km. This includes upper-air parameters on 37 fixed pressure levels from 1,000 to 1 hPa and 137 model levels distributed using hybrid sigma-pressure coordinate system.. For this study, we interpolate ERA5 400 hPa fixed pressure level data to assess atmospheric water vapor conditions near Mount Ruang concurrent with FY-3E/HIRAS-II observations.

## 3 Channel selection method

When selecting channels, it is crucial to avoid bands with cloud or aerosol interference and long-wave channels that provide redundant information (Tsuchiya, 1983). In addition, as the temperature Jacobian matrices of the water vapor and ozone channels can be strongly influenced by the state of the atmosphere, they should not be used as the main sources of temperature information (Kuai et al., 2010). Therefore, different sets of channels should be considered at various stages during the channel selection process. This research suggests two primary steps for channel selection, as follows.

1. Initially, channels are excluded through pre-screening, which eliminates regions of high uncertainty in the simulated spectrum based on specific criteria.

2. The primary channel selection algorithm is based on Jacobian calculations as a measure of the information content of various atmospheric species and is executed through multiple independent selection operations.

### 3.1 Channel pre-screening

Channel pre-screening rejects spectral regions that would bring substantial uncertainty in the subsequent simulation phase, thus enhancing the efficiency of and reducing data redundancy in the forward simulations (Li et al., 2022). We pre-screened the mid- and long-wave IR bands by eliminating trace gas absorption channels and applying a threshold to the noise equivalent target brightness temperature (BT) difference (NEdT).

The first step eliminates channels with strong absorption of trace gases. For any of the six standard atmospheric profiles, channels are removed if changes in trace gas content induce a BT shift of >1 K. Channels are retained if the gas-induced BT change is <1 K; the influence of these gases is then incorporated into the forward model for simulation. Among nine trace gases ($CH_4$, CO, $N_2O$, $CCl_4$, CFC-11, CFC-12, CFC-14, $HNO_3$, $NO_2$, OCS, and NO), only the first three significantly affect the channel BT (Collard, 2007). As the absorption bands of CO and $N_2O$ fall outside this study's spectral range, we focus on $CH_4$ for testing. Channels significantly influenced by ozone and solar irradiance are also excluded.

The second step involves eliminating channels with excessive noise. To minimize the risk of excluding relevant spectral bands or retaining inappropriate bands, a threshold of 0.2 K for NEdT is adopted as the pre-screening criterion for channel selection.

The third step excludes channels with non-linear Jacobian matrix and multiple Jacobian peaks. Using the LBLRTM model and six standard atmospheric profiles, we calculate the Jacobian matrix for temperature and water vapor. Channels exhibiting significant double or multiple peaks in the Jacobian matrix are excluded. Figure 2 illustrates the channels rejected during pre-screening: the red areas indicate channels influenced by $O_3$, purple areas are those affected by $CH_4$, and yellow areas those with multiple peaks in the Jacobian matrix.

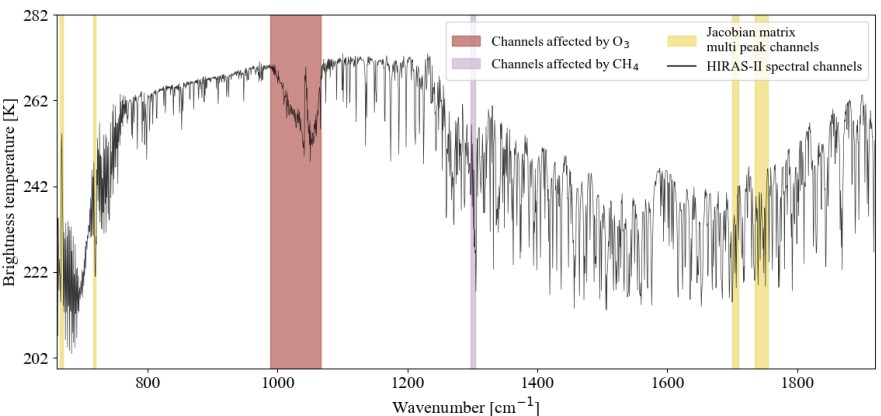

**Figure 2: FY-3E/HIRS-II channel pre-screening results: red and purple highlight channels affected by $O_3$ and $CH_4$, respectively; yellow highlights channels with multiple peaks in the Jacobian matrix.**

### 3.2 Jacobian matrix based information analysis

We calculate and analyze the information generated by water vapor, temperature, and $SO_2$ at different altitudes to select and utilize the most relevant channels. To evaluate the capability of HIRAS-II channels to provide information on these parameters,

we employ the Jacobian matrix for channel selection. The Jacobian functions can identify a set of optimal channels with maximum or minimum information content for each atmospheric profile. It assesses the sensitivity of radiation to the specific physical and chemical parameters. For a specified wavenumber ($v$), the sensitivity of BT to variations in geophysical parameters ($X$) is represented by the Jacobian matrix for each pressure layer (Coopmann et al., 2020) as follows:

$$J_v(X) = \frac{\partial BT(v)}{\partial X},$$

(2)

The Jacobian matrix illustrates the sensitivity of atmospheric BT to temperature, humidity, and various gas concentrations at a given wavenumber (Aires et al., 2016).

Three key parameters for measuring the properties of a Jacobian matrix are employed. The first parameter is the maximum value of each Jacobian matrix, denoted as $M$, quantifies the information (here, all discussions of $M$ in this paper only consider its maximum value, i.e., $|M|$). The second is the pressure level $P$ corresponds to the height where the Jacobian matrix attains its peak value, indicating the altitude at which the IR radiation is the most responsive to variations in atmospheric composition. The third parameter, dP, represents the width at half maximum of the Jacobian matrix peak, defined as the pressure difference between the two levels where the Jacobian matrix value drops to half of its maximum. This metric represents the vertical extent of the atmospheric layer contributing most significantly to the IR signal. Figure 3 schematically represents the SO₂ profile, the Jacobian peak and the maximum half-width of the Jacobian function under the conditions of the US standard atmosphere, 1976.

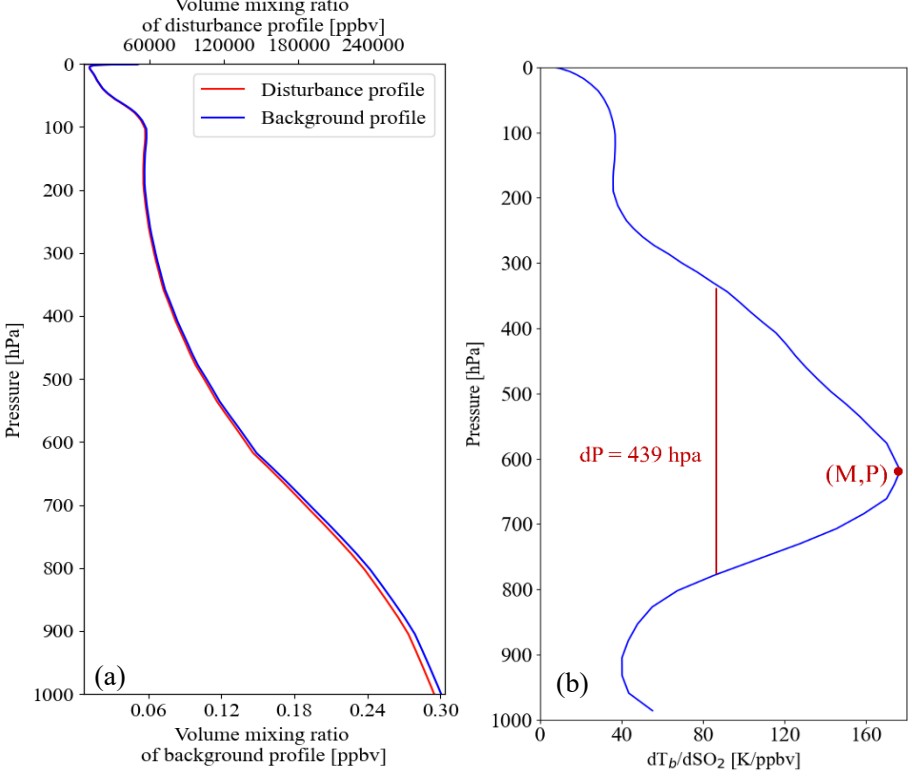

**Figure 3: Representation of the maximum half-width and peak value of the SO₂ Jacobian function for the US Standard Atmosphere, 1976: (a) SO₂ profile, (b) 1163.125 cm⁻¹ channel.**

To accurately monitor $SO_2$, it is essential to minimize the interference of atmospheric temperature and water vapor on the $SO_2$ channels. Since the radiance signals from $SO_2$ channels are simultaneously influenced by atmospheric temperature, water vapor, and $SO_2$, it is necessary to utilize other channels to provide independent atmospheric temperature and water vapor information for separation. In selecting channels minimally influenced by atmospheric temperature, we prioritize those channels that are primarily sensitive to a single gas with a constant concentration, $CO_2$ absorption channels primarily reflect the information of atmospheric temperature profiles (Li et al., 2022). Consequently, we utilize the spectral absorption region of $CO_2$ (666 – 1000 cm⁻¹) to calculate the temperature Jacobian matrix and combine this with the atmospheric IR window channel to select the atmospheric temperature channels. Water vapor channels contain both temperature and water vapor information, while $SO_2$ channels contain information of temperature, water vapor and $SO_2$. To separate temperature from water vapor in water vapor absorption channel radiances, $CO_2$ channels play an important role through providing temperature information. If a water vapor absorption channel and a $CO_2$ absorption channel have similar temperature Jacobian, they have also similar temperature sensitivity, and thus that $CO_2$ channel is helpful for separating the temperature from water vapor in the water vapor channel radiance. Same for a $SO_2$ channel, if a water vapor channel has similar temperature Jacobian and water vapor Jacobian, then the water vapor channel is helpful for separating temperature and water vapor from $SO_2$ in that $SO_2$ channel radiance. During the cross-comparison of channel selection, we ensure that the water vapor Jacobian matrix and temperature Jacobian matrix within the water vapor absorption region are consistent with those in the $SO_2$ channels. Thus, when subtracting the brightness temperature of the $SO_2$ channels from that of the water vapor channels, the influence of water vapor, atmospheric temperature, and surface radiation shared by both channels can be effectively removed.

The specific channel selection process is shown in Fig. 4, it illustrates the cross-comparison process using the three key parameters of Jacobian matrices in the range of $SO_2$, water vapor and $CO_2$ absorption regions. Initially, we computed the temperature, water vapor, and $SO_2$ Jacobian matrix for the six standard atmospheric profiles. Then, the similarity in the peak and half-width of the Jacobian matrix at specific pressure level P for HIRAS-II channels in $SO_2$, water vapor and temperature absorption region were cross-compared. The temperature Jacobian information for the atmospheric temperature channels and the water vapor Jacobian information for the water vapor channels needs to align with that for the $SO_2$ channels to minimize the influence of atmospheric water vapor and temperature to $SO_2$. Similarly, the temperature and water vapor Jacobian information for the water vapor channels must match the corresponding information for the $SO_2$ channels. Consequently, when $SO_2$ concentration changes, the similarity of the water vapor and temperature Jacobian matrices between the $SO_2$ channels and the water vapor channels can effectively eliminate the interference of atmospheric temperature  and water vapor on $SO_2$ monitoring results.

Using this information, we then identified the atmospheric temperature channels, water vapor absorption channels, and $SO_2$-sensitive channels. Considering the variability in the sensitivity of the HIRAS-II channels to the atmospheric conditions, we

utilize 1040 hPa as the near-surface atmospheric pressure and compute the Jacobian matrices for water vapor, temperature, and $SO_2$ across 99 vertical atmospheric sections of the six atmospheric profiles.

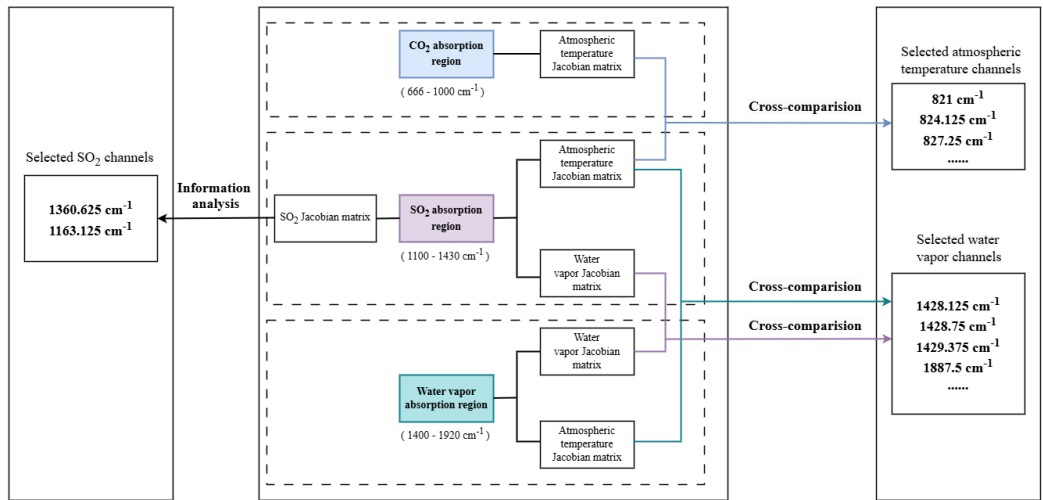

**Figure 4: Schematic diagram of channel selection method.**

### 245  3.2.1 SO₂ channel selection

In situ measurements reported by Rose et al. (2004) indicate $SO_2$ concentrations of 500 – 1000 ppbv during an aircraft encounter with a 35-hour-old volcanic plume from the Icelandic Hekla eruption in February 2000, at a distance of approximately 1300 km from the source. In comparison, the concentration of $SO_2$ in the clean troposphere typically ranges from 0.25 – 0.43 ppbv (Casadevall et al., 1984). Given that $SO_2$ concentrations increase dramatically over a short period during

volcanic eruptions, for $SO_2$, we perturb the atmospheric profiles at different pressure levels using $5 \times 10^4$ times gas content, to better represent the gas distribution characteristics in volcanic eruption scenarios. Given the low $SO_2$ content under the other five atmospheric conditions, this study focuses on the $SO_2$ information for the US Standard Atmosphere, 1976. The corresponding $SO_2$ Jacobian functions (Fig. 5) clearly shows that the $SO_2$ absorption region is located mainly around the central wavenumbers of 1360 and 1163 cm$^{-1}$. The 1360 cm$^{-1}$ band exhibits the strongest $SO_2$ signal among the available spectral

bands. However, it is also a strong absorption region for atmospheric water vapor, which can introduce contamination in $SO_2$ retrievals. This band demonstrates minimal sensitivity to radiative contributions from the surface and lower atmosphere, making it particularly effective for monitoring stratospheric $SO_2$ plumes (Thomas & Watson, 2010). In contrast, the 1163 cm$^{-1}$ band falls within an atmospheric window region. While the presence of $SO_2$ in this band leads to a certain degree of radiative attenuation, it remains well-suited for detecting $SO_2$ plumes in the troposphere (Carboni et al., 2016). This characteristic makes

it especially valuable for monitoring volcanic activity characterized by continuous passive degassing. By leveraging the complementary strengths of these bands, we select $SO_2$-sensitive channels with a central wavenumber around 1163 and 1360 cm$^{-1}$. In addition, $SO_2$ absorption information is discernible at various altitudes in the atmosphere, particularly in the

middle atmosphere and near the surface. To obtain pure SO₂ absorption information, it is essential to eliminate information about the surface temperature, atmospheric temperature, and water vapor that might interfere with the SO₂ observation channels,

thereby avoiding overestimation or misestimation of the SO₂ content and dispersion trends. We selected the top channels with the highest Jacobian matrix values in the SO₂ absorption region near 1360 and 1163 $cm^{-1}$, which are 1360.625, and 1163.125 $cm^{-1}$ . These two channels contain prominent SO₂ absorption information.

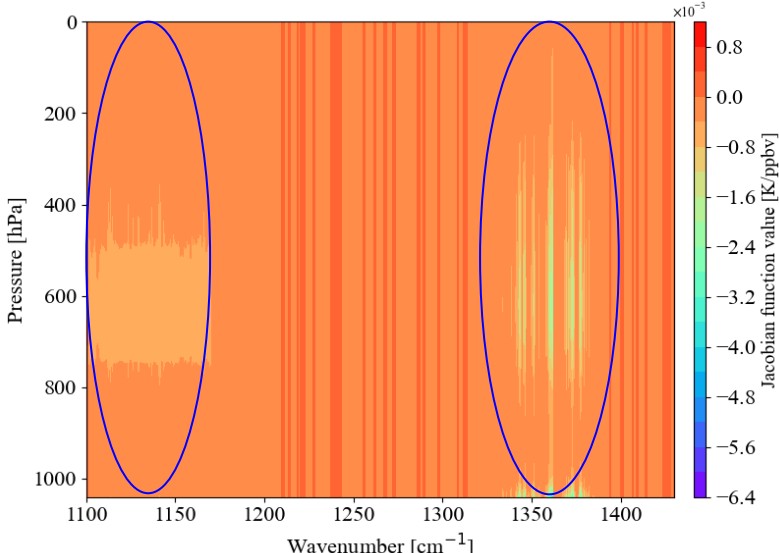

**Figure 5: Schematic diagram of the SO₂ Jacobian matrix with atmospheric profiles from the US Standard Atmosphere, 1976.**

### 3.2.2 Atmospheric temperature channel selection

Volcanic eruptions typically change the temperature of the stratosphere and troposphere, making it essential to eliminate any

interference effect of atmospheric temperature on SO₂ observations (Yang & Schlesinger, 2002). Figure 6(a)-(f) shows temperature Jacobian functions for the six atmospheric profiles, revealing that near-surface temperatures are more responsive to temperature perturbations in the tropical, mid-latitude summer, subarctic summer, and US Standard Atmosphere, 1976, profiles, while the mid-latitude winter and subarctic winter profiles exhibit greater fluctuations at higher altitudes. For the atmospheric temperature channels, it is crucial that the temperature Jacobian functions peak at the same altitude as those of

the SO₂ channels and have similar half-widths of their Jacobian functions. We compare the temperature Jacobian functions of the SO₂ channels with that of the atmospheric temperature absorption region under each set of atmospheric profiles, so that each channel in the atmospheric temperature absorption region can be compared with all channels in the SO₂ absorption region for atmospheric temperature absorption information. First, we filter out channels where both peak at the same altitude. Then we determine the final atmospheric temperature channels using a threshold of the half-width difference being <0.1. Channels

meeting these conditions, along with the SO₂ channels, exhibit consistent temperature absorption information and adequately

cover the atmospheric temperature channels for the six observed atmospheric conditions. According to Fig. 7, many channels in atmospheric temperature absorption region also have similar atmospheric temperature absorption information with multiple SO₂ channels at the same time.

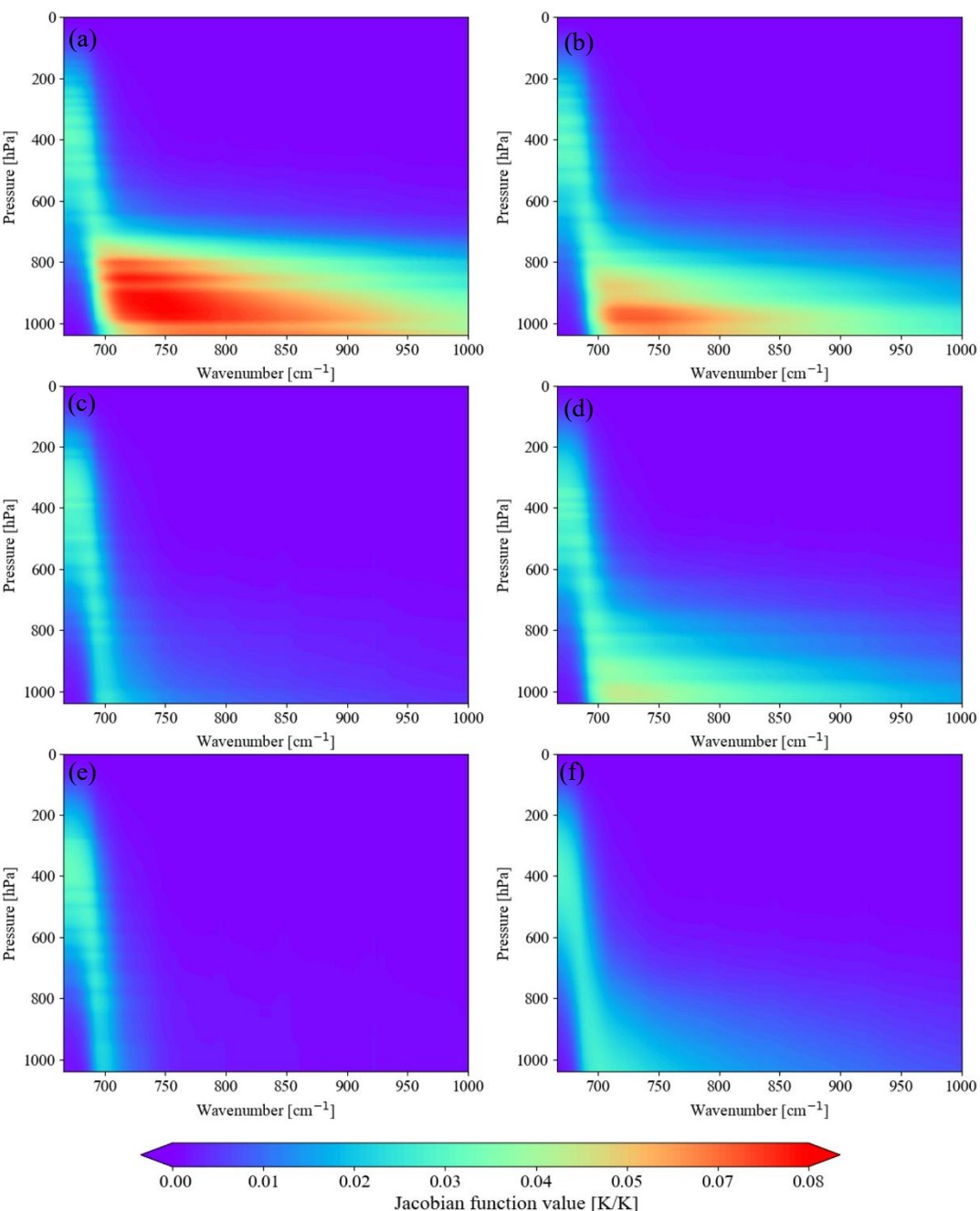

**Figure 6: Representations of temperature Jacobian functions at atmospheric temperature absorption region for the conditions of six atmospheric profiles: (a) tropical atmospheric profile, (b) mid-latitude summer atmospheric profile, (c) mid-latitude winter atmospheric profile, (d) subarctic summer atmospheric profile, (e) subarctic winter atmospheric profile, and (f) US Standard Atmosphere, 1976.**

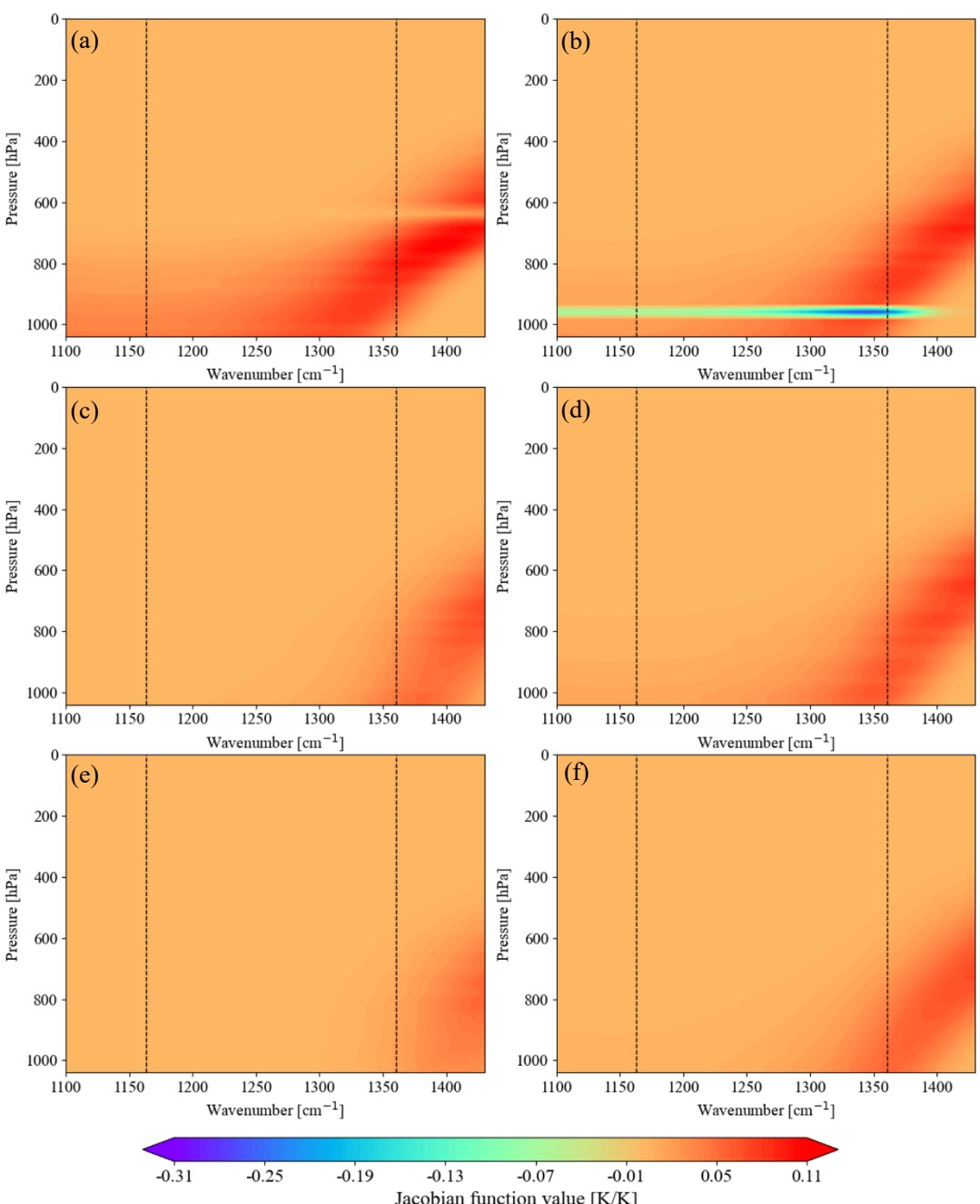

**Figure 7: Representations of temperature Jacobian functions at SO₂ absorption region (black dashed lines represent selected SO₂ channels) for the conditions of six atmospheric profiles: (a) tropical atmospheric profile, (b) mid-latitude summer atmospheric profile, (c) mid-latitude winter atmospheric profile, (d) subarctic summer atmospheric profile, (e) subarctic winter atmospheric profile, and (f) US Standard Atmosphere, 1976.**

### 3.2.3 Water vapor absorption channel selection

Figure 8 shows strong absorption by water vapor around 1428 and 1850 cm⁻¹ under the six atmospheric conditions, indicating this region contains substantial absorption information on water vapor. In addition, the absolute value of the Jacobian function for water vapor in the lower and middle layers of the 1428 cm⁻¹ band can reach up to $-9.7 \times 10^3$ K/ppbv under the tropical, meanwhile, mid-latitude summer, subarctic summer, and 1976 US Standard Atmosphere profiles, indicating that water vapor has a stronger influence than in the mid-latitude winter and subarctic winter profiles. At the same time, it can be seen from Fig.

9 that the SO₂ absorption region around 1360 cm⁻¹ is more susceptible to water vapor contamination than the 1163 cm⁻¹ absorption region. Under most atmospheric profile conditions, there exists a channel within the water vapor absorption region that exhibits Jacobian characteristics consistent with the selected SO₂ channels according to Fig. 9. We calculate the temperature Jacobian functions and water vapor Jacobian functions separately within the water vapor absorption region and SO₂ absorption region. The Jacobian information of water vapor in SO₂ and water vapor absorption region are cross compared.

The Jacobian information of atmospheric temperature in SO₂, water vapor absorption region and selected atmospheric temperature channels are also cross compared and the channels with consistent maximum peak value and half-width were selected to ensure that the vertical changes of water vapor and atmospheric temperature were consistent with those of SO₂. The cross-comparison criteria of the Jacobian matrix here are consistent with the selection criteria and threshold of the atmospheric temperature channels in section 3.2.2. Through the cross-comparison process, the selected water vapor channels

can simultaneously contain consistent atmospheric temperature and water vapor absorption information to the SO₂ channels. In this way, the atmospheric temperature and water vapor absorption information carried in the selected SO₂ channels can be removed in the subsequent calculation of the BT difference between the SO₂ channels and the water vapor channels. Figure 10 illustrates the specific central wavenumbers of the selected atmospheric temperature channels, water vapor absorption channels and their corresponding BTs under the 1976 US Standard Atmosphere.

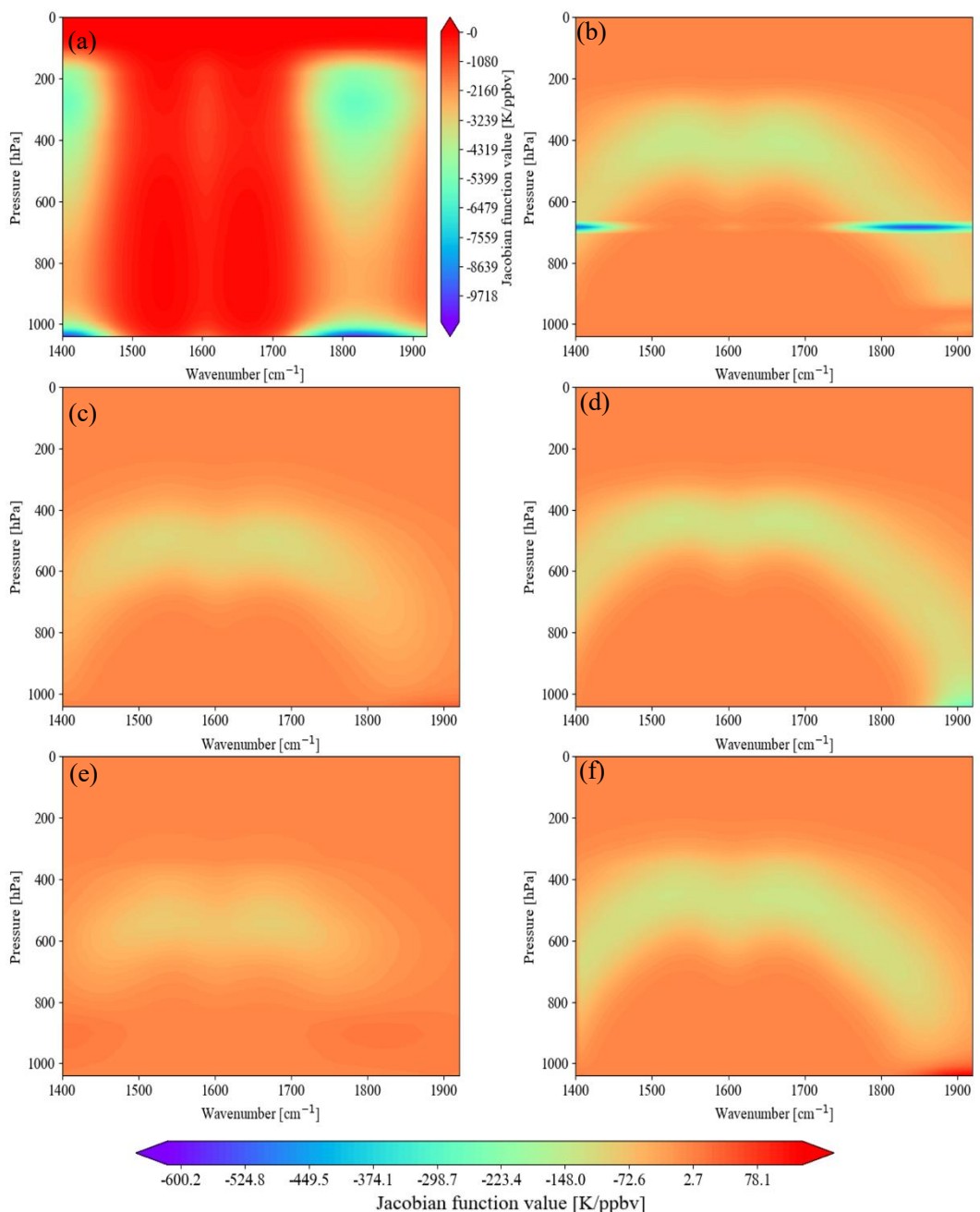

**Figure 8: Representations of water vapor Jacobian functions at water absorption region for conditions of six atmospheric profiles: (a) tropical atmospheric profile, (b) mid-latitude summer atmospheric profile, (c) mid-latitude winter atmospheric profile, (d) subarctic summer atmospheric profile, (e) subarctic winter atmospheric profile, and (f) US Standard Atmosphere, 1976.**

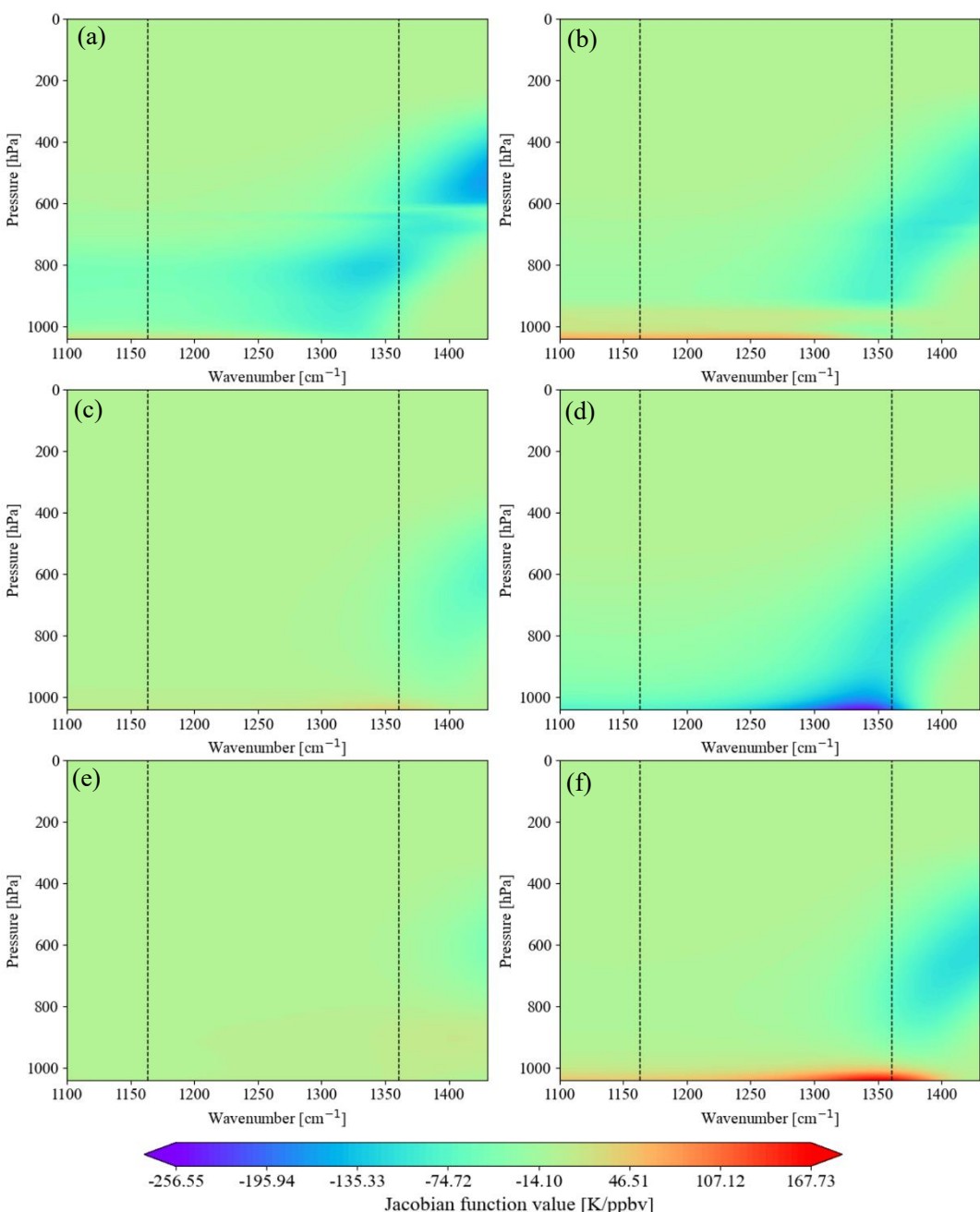

**Figure 9: Representations of water vapor Jacobian functions at SO₂ absorption region (black dashed lines represent selected SO₂ channels) for conditions of six atmospheric profiles: (a) tropical atmospheric profile, (b) mid-latitude summer atmospheric profile, (c) mid-latitude winter atmospheric profile, (d) subarctic summer atmospheric profile, (e) subarctic winter atmospheric profile, and (f) US Standard Atmosphere, 1976.**

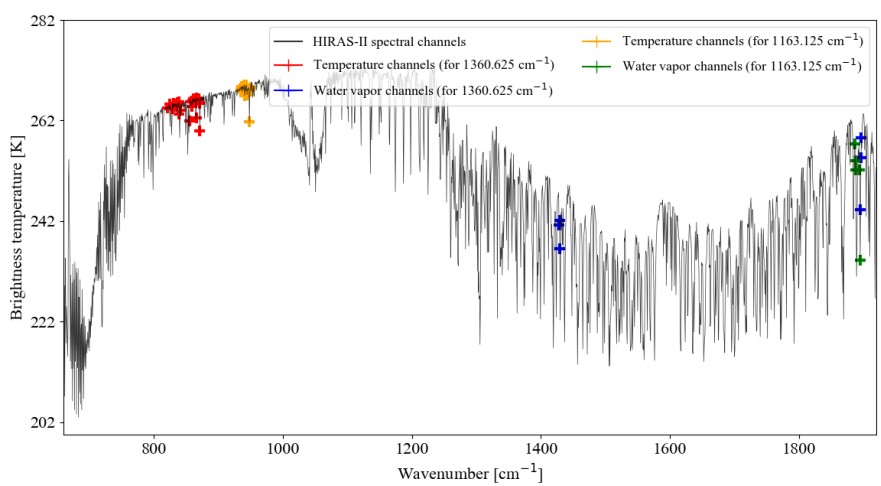

**Figure 10: Part of the HIRAS-II brightness temperature spectrum with selected atmospheric temperature channels and water vapor absorption channels labelled.**

Under the same $SO_2$ and water vapor conditions and based on the selected $SO_2$ channels, we respectively selected three corresponding water vapor channels for both the 1163.125 and 1360.625 cm$^{-1}$ channels whose channel combination with the largest brightness temperature difference. By analyzing the BT difference, we determined the $SO_2$ sensitive channels to accurately carry out the $SO_2$ retrieval. As can be seen in Fig. 11, 1163.125 and 1360.625 cm$^{-1}$ is used as the $SO_2$-sensitive channels, 1887.5 and 1429.375 cm$^{-1}$ as the water vapor absorption channels. For 1360.625 cm$^{-1}$ channel, the combination of

the channels we chose can effectively remove the water vapor information contained in the $SO_2$-sensitive channels and can also better demonstrate the $SO_2$ plume after deducting the effect of water vapor, which lays the foundation for the $SO_2$ retrieval in the subsequent inversion process.

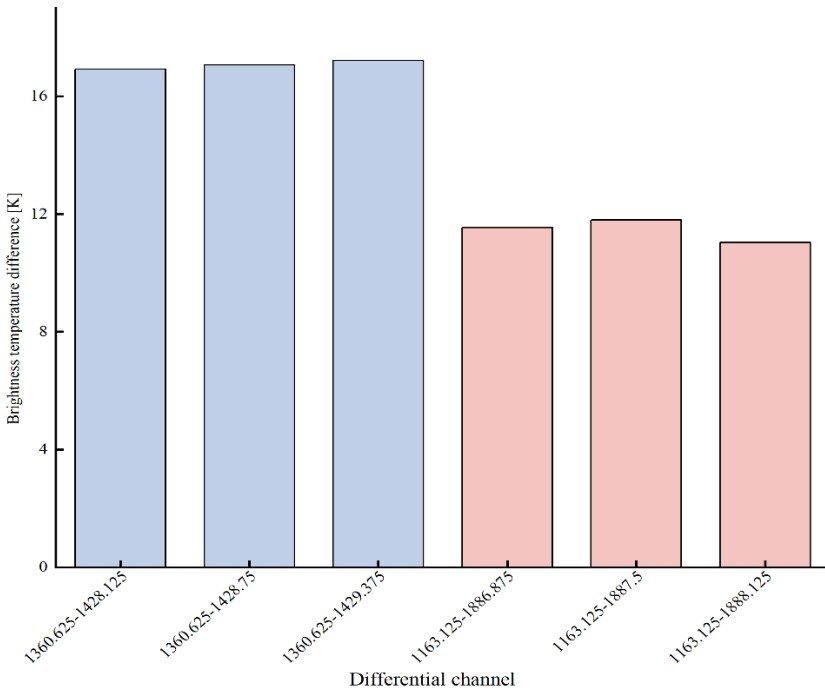

**Figure 11: Brightness temperature difference between SO₂ channel and water vapor absorption channel with atmospheric profiles from the 1976 US Standard Atmosphere.**

### 3.3 Surface temperature channel selection

Land surface temperature (or surface skin temperature) is a key variable in IR data inversion (Jimenez-Munoz et al., 2009). The atmosphere minimally reflects, scatters, and absorbs electromagnetic waves in the atmospheric IR window band (Senf & Deneke, 2017). Therefore, we select the clean channel from this range with the highest BT: its use in subsequent analyses as the land surface temperature channels mitigates the influence of land on SO₂ observations. Table 2 presents the distribution of the three channels with the highest BT across the six atmospheric profiles. Notably, the land surface temperature channels for

the mid-latitude winter and subarctic winter situations are identical, while those for the mid-latitude summer and subarctic summer profiles are somewhat similar. The tropical atmosphere profile has a land surface temperature channel with a higher wavenumber and shorter wavelength compared with the other profiles. The land surface temperature channel for the US Standard Atmosphere, 1976, falls between those of the other profiles. To ensure the selected land surface temperature channels are applicable to most atmospheric conditions, we identify the two channels with the highest frequency (902.5 and

901.875 cm$^{-1}$) for subsequent work.

Table 2 Distribution of surface temperature channels under six atmospheric profiles

| Atmosphere profile | Channel wavenumber (cm$^{-1}$) | | |
|---|---|---|---|
| Tropical | 916.875 | 905.625 | 906.875 |
| Midlat Summer | 904.375 | 903.75 | 902.5 |
| Midlat Winter | 901.25 | 901.875 | 902.5 |
| Subarctic Summer | 904.375 | 901.875 | 902.5 |
| Subarctic Winter | 901.25 | 901.875 | 902.5 |
| US1976 | 901.25 | 901.875 | 902.5 |

## 4 Sensitivity analysis

### 4.1 Effects of differences in surface temperature and near-surface atmospheric temperature on SO₂-sensitive channels

Given the variations in surface characteristics affecting atmospheric radiation, we analyzed the impact of the generally low temperature difference between the surface and the overlying air on the SO₂ Jacobian function. Meanwhile, the $750 - 1200$ cm$^{-1}$ region is highly sensitive to surface features (Clarisse et al., 2010), and the sensitivity of HIRAS-II to SO₂ is significantly influenced by the temperature difference (TD) between the surface and the first distinct layer of air ($T_p$) (Tsuchiya, 1983). The Jacobian formula defines the relationship between the change in brightness temperature and the perturbation in material concentration. Under consistent atmospheric conditions with fixed SO₂ concentration perturbations and uniform background brightness temperature, the TD after SO₂ perturbation demonstrates a similar trend and behavior to that of the Jacobian value. As a result, TD can effectively substitute for the Jacobian value in assessing the detection capability of SO₂. For simplicity, we consider three scenarios: $T_s = T_p$ (TD = 0), $T_p > T_s$ (TD > 0), and $T_p < T_s$ (TD < 0). With $\varepsilon = 0.98$ and $P = 212$ hPa, TD was varied from $-10$ to 10 K in 5 K increments, and infrared radiation was simulated under each set of conditions. Figure 12 illustrates variations in the SO₂ plume in 1163.125 and 1360.625 cm$^{-1}$ channels under different TD conditions for the US Standard Atmosphere, 1976.

From Fig. 12(a), it can be observed that for the 1360.625 cm$^{-1}$ channel, SO₂ with column densities <150 DU exhibits high sensitivity to changes in the TD. However, when the SO₂ column density >150 DU, the response of TD to concentration variations significantly weakens, indicating that this channel tends to saturate at higher SO₂ concentrations. This phenomenon demonstrates that the 1360.625 cm$^{-1}$ channel is more effective for detecting SO₂ in the middle and upper troposphere. In contrast, as shown in Fig. 12(b), for the 1163.125 cm$^{-1}$ channel, a positive change in TD leads to a significant increase in brightness temperature at the same SO₂ concentration. As the SO₂ concentration increases, the influence of TD on brightness temperature decreases approximately linearly. This suggests that the 1163.125 cm$^{-1}$ channel is more susceptible to interference

from surface and near-surface radiation properties, with its signal primarily reflecting the distribution of SO₂ in the lower atmosphere.

For a plume SO₂ content of <150 DU, an increasingly positive TD enhances SO₂ detection in the IR band. Conversely, a decrease in TD limits SO₂'s contribution to radiation, thereby constraining its IR remote sensing capability. As the plume's SO₂ content increases, the impact of TD on SO₂ observation diminishes. These findings suggest that favorable TD conditions

can enhance the accuracy of SO₂ detection and inversion, which is relevant to monitoring air quality. Due to the vertical distribution of gases, near-surface SO₂ tends to be underestimated, but a positive TD helps capture the net absorption of near-surface SO₂.

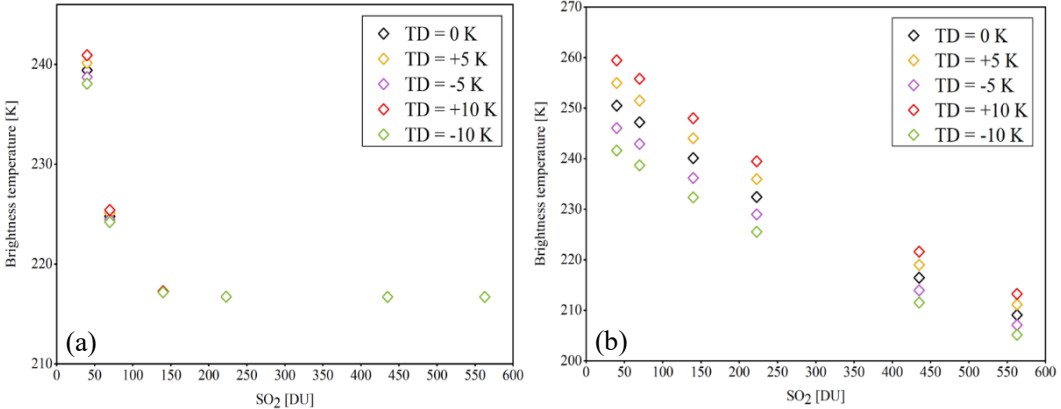

**Figure 12: Sensitivity of SO₂ plume measurement at channels (a) 1360.625 and (b) 1163.125 cm⁻¹ to surface temperature with atmospheric profiles from the US Standard Atmosphere, 1976.**

### 4.2 SO₂ plume sensitivity

This study assumes an atmosphere containing SO₂ clouds at various altitudes and simulates the radiative transfer in a standard

atmosphere with an introduced SO₂ layer of varying SO₂ concentration. The simulations replicate FY-3E/HIRAS-II's observations of SO₂ volcanic plumes, focusing on the sensitivity of the differences in BT between central wavenumbers of 1360.625 and 902.5 cm⁻¹ and between 1163.125 and 902.5 cm⁻¹ to the total SO₂ column in Dobson units at four plume altitudes (3, 6, 12, and 16 km). The temperature and humidity profiles for these simulations are based on the US Standard Atmosphere, 1976. Figure 13(a) shows that for SO₂ plumes under varying pressure intensities, strong sensitivity is observed when SO₂

content exceeds 50 DU. At 50 ~ 300 DU, the sensitivity of the SO₂ plume increases with altitude. However, beyond 300 DU, the impact of altitude on sensitivity diminishes, indicating a saturation state. Thus, the 1360.625 cm⁻¹ channel is prone to saturation at high SO₂ concentrations. Figure 13(b) shows that for SO₂ plumes below 400 DU, the SO₂ Jacobian function value for the 1163.125 cm⁻¹ channel is relatively low, resulting in reduced sensitivity. Conversely, above 500 DU, the channel exhibits a more pronounced response to increasing SO₂ concentration and plume height.

Therefore, combining these two channels for different SO₂ concentrations enables the representation of a broad range of net SO₂ absorption. The brightness temperature difference between the 1360.625 and 902.5 cm$^{-1}$ channels can reach up to ~70 K, aligning well with previous experimental results (Ackerman et al., 2008).

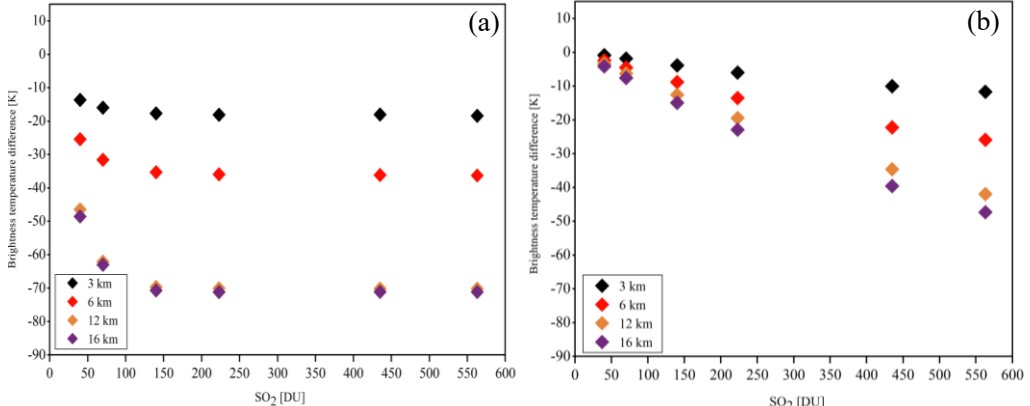

**Figure 13: Modelled FY-3E/HIRAS-II brightness temperature differences between the (a) 1360.625 and 902.5 cm$^{-1}$ channels and the (b) 1163.125 and 902.5 cm$^{-1}$ channels for assessing column SO₂ content (DU) at four plume heights in atmospheric profiles derived from the US Standard Atmosphere, 1976.**

## 5 Case study

The channels for SO₂ detection and retrieval least affected by temperature and water vapor were selected based on experimental
results. To verify the accuracy of our channel selection, we compared observations of a volcanic eruption using our selected channels and normal channels.

The selected eruption was of Mount Ruang, Indonesia, the southernmost complex volcano in the Sandwich Islands. Its first recorded eruption in 1808 forced the evacuation of over 1,000 people (Galetto et al., 2024). Its violent eruption on the evening of 17 April 2024 was observed by FY-3E/HIRAS-II on 18 April. The collected data are used to explore the advantages of our
selected channels.

Figure 14 depicts the differences between the following pairs of channels: 1360.625 and 902.5 cm$^{-1}$, 1360.625 and 1429.375 cm$^{-1}$, 1163.125 and 902.5 cm$^{-1}$, and 1163.125 and 1887.5 cm$^{-1}$. Comparison of the difference results of Fig. 14(a) and Fig. 14(b) indicates that the extent of the SO₂ plume near the volcano's center may be mistaken for water vapor due to the background channel's inability to effectively remove the effect of water vapor from the 1360.625 cm$^{-1}$ channel. Water vapor
far from the crater is prone to misclassification as SO₂ gas. A comparison of the Fig. 14(c) and Fig. 14(d) sets of difference results indicates that it is challenging to separate SO₂ from the atmosphere due to the smaller value of the SO₂ Jacobian matrix for the 1163.125 cm$^{-1}$ channel and its lower sensitivity to SO₂ information compared with the 1360.625 cm$^{-1}$ channel. In addition, the eruption increased the atmospheric temperature near the volcano, and the difference between the 1163.125 and 1887.5 cm$^{-1}$ channels cannot remove the atmospheric temperature information observed by the sensors, resulting in significant

BT differences over a large area, compared to the former, the difference between the 1163.125 and 902.5 cm$^{-1}$ channels allows for a more pronounced enhancement of certain SO$_2$ plumes, but the results were still suboptimal. Figure 14(b) shows the BT difference between the most sensitive and background channels based on the experimental selection. The chosen combination of SO$_2$ channels filters out most of the water vapor and atmospheric temperature effects in the observation channel, resulting in better detection of small SO$_2$ plumes.

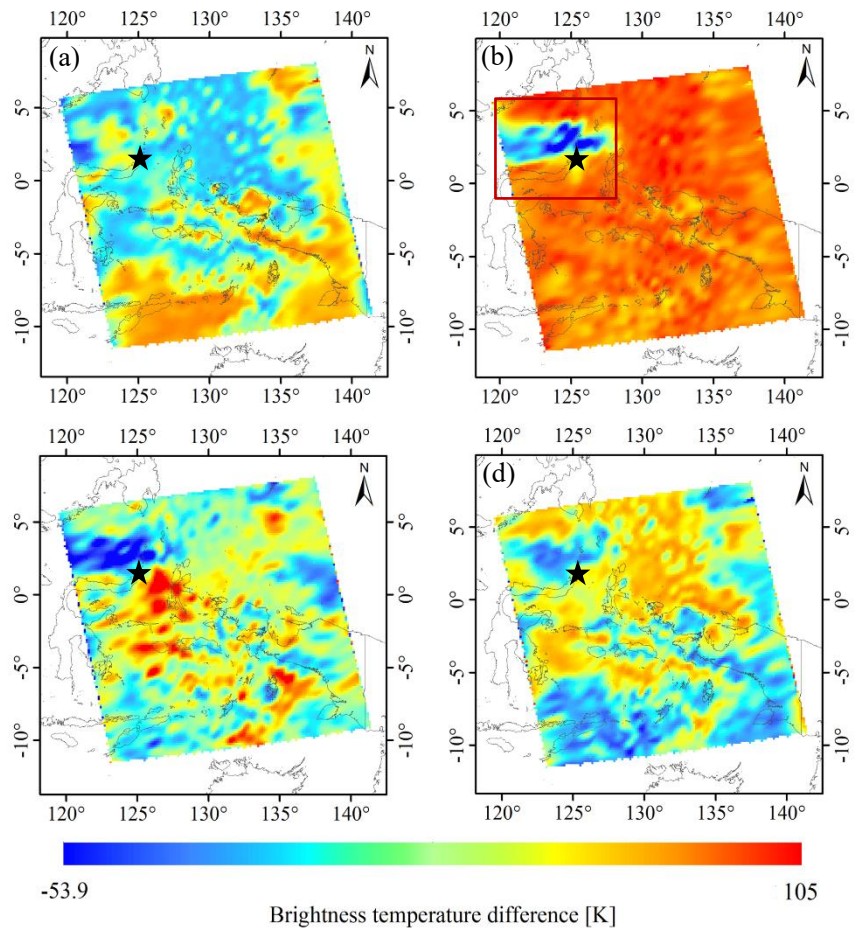

**Figure 14: FY-3E/HIRAS-II brightness temperature difference data for the region around Mount Ruang (black star in each image) at 08:55 UT on 18 April 2024 for the channels (a) 1360.625 and 902.5 cm$^{-1}$, and (b) 1360.625 and 1429.375 cm$^{-1}$, (c) 1163.125 and 902.5 cm$^{-1}$ and (d) 1163.125 and 1887.5 cm$^{-1}$.**

Figure 15 compares the FY-3E/HIRAS-II BT difference data (for the area indicated by the red box in Fig. 14(b)) with corresponding observations by Sentinel-5P/TROPOMI. The area of the SO$_2$ plume's spread and its trajectory are essentially the same for both cases. Figure 16 shows the absolute humidity data at 09:00 UT on 18 April 2024 from the ERA5 atmospheric reanalysis data at an atmospheric pressure of 400 hPa, confirming that the SO$_2$ plume observed by FY-3E/HIRAS-II in Fig. 14 is largely free of interference by water vapor.

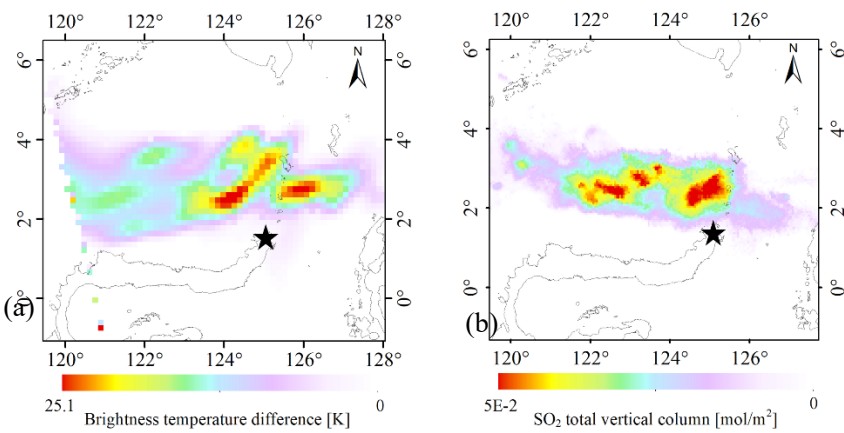

**Figure 15: Comparison of SO₂ around Mount Ruang (black star in each image) observed by FY3E/HIRAS-II on 18 April at 08:55 UT and Sentinel-5P/TROPOMI on 18 April at 04:07:08 UT.**

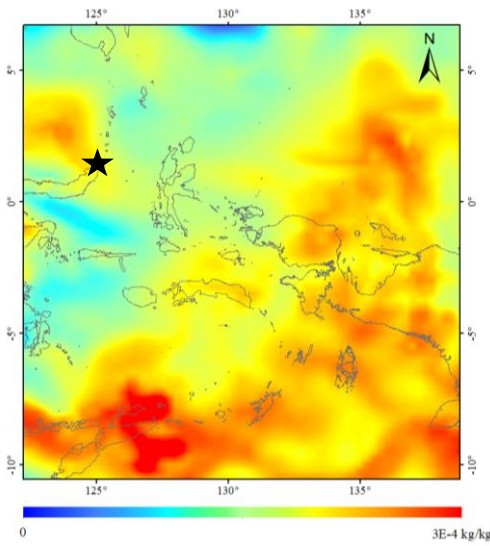

**Figure 16: Specific humidity data from ERA5 for the area around Mount Ruang (black star) at 09:00 UT on 18 April 2024 at an atmospheric pressure of 400 hPa.**

## 6 Summary and conclusion

This paper proposes a novel methodology for selecting SO₂ sensitive channels from FY-3E/HIRAS-II hyperspectral IR atmospheric sensors to quantitatively monitor volcanic SO₂. The peak and maximum half-width of the Jacobian function of

SO$_2$, temperature and water vapor under different atmospheric conditions were cross compared to identify the optimal channels for SO$_2$ detection and retrieval. The results demonstrate that the 1360.625 cm$^{-1}$ channel (wavelength around 7.3 μm) is most

sensitive to SO$_2$, exhibiting a maximum peak and half width Jacobian values that conveys comprehensive SO$_2$ absorption information. While 1163.125 cm$^{-1}$ (wavelength around 8.6 μm) channel has a weaker absorption to SO$_2$ compared to 1360.625 cm$^{-1}$ channel but also contains valuable information.

Through cross-comparison of the Jacobian matrices of water vapor, temperature and SO$_2$, it is found that the 1429.375 cm$^{-1}$ channel (wavelength around 7.0 μm) can not only reflect the water vapor information to the greatest extent, but also maintain

consistent variations with the atmospheric temperature and SO$_2$, which allows to minimize the influence of atmospheric water vapor and temperature on SO$_2$ detection and retrieval. In the atmospheric IR window band, we identify two channels (902.5 and 901.875 cm$^{-1}$) with the highest frequency of maximum BT under different atmospheric conditions as the land surface temperature channel to mitigate the influence of land on SO$_2$ observations.

A sensitivity study shows that the BT difference (BTD) between the experimentally selected SO$_2$ sensitive channel

(1360.625 cm$^{-1}$ channel) and the background channel (902.5 cm$^{-1}$ channel) demonstrates a pronounced relationship to SO$_2$ at 50 ∼ 300 DU. To address the phenomenon of saturation of the SO$_2$ response in the 1360.625 cm$^{-1}$ channel at high concentrations, we propose to use the 1163.125 cm$^{-1}$ channel to provide auxiliary information. It is demonstrated that the 1163.125 cm$^{-1}$ channel exhibits a more significant and linear response to increasing SO$_2$ concentration and plume height when the SO$_2$ is above 500 DU. In addition, in the lower and middle layers, a positive difference between the surface air temperature

and the surface skin temperature enables the IR band to capture more SO$_2$ information. By further analyzing the BTD between the 1360.625 cm$^{-1}$ and 1429.375 cm$^{-1}$, the influence of water vapor and atmospheric temperature from 1360.625 cm$^{-1}$ can be effectively removed.

The main advantage of this methodology is that it comprehensively considers the interference of atmospheric temperature, humidity, and surface temperature on SO$_2$ detection and retrieval, laying the groundwork for developing a more accurate and

flexible volcanic SO$_2$ retrieval algorithm under different atmospheric conditions. Traditional broadband multispectral satellites are seriously influenced by water vapor and atmospheric temperature in SO$_2$ absorption region, and it is difficult to accurately separate water vapor and temperature information from SO$_2$ sensitive channels. This methodology overcomes the above problem using satellite-based hyperspectral IR data under a Jacobian Matrix information framework. This method is able to greatly enhance the efficiency for extracting SO$_2$ information from hyperspectral IR sounder with large number channels while

maintain the accuracy. Therefore, it has great potential in both satellite-based and ground-based hyperspectral data processing for volcanic SO$_2$ retrieval.

For future work, development of a comprehensive dataset representing a variety of volcanic ash spectral properties and atmospheric conditions for SO$_2$ modeling, detection, and retrieval, is highly desired. Building on the dataset and the traditional line by line forward radiation transfer model, machine learning methods can help explore the nonlinear relationship between

volcanic SO$_2$ and the atmosphere/surface signals from massive forward simulated samples, as well as develop a fast and accuracy radiative transfer model for SO$_2$ retrieval.

## CRediT authorship contribution statement

**Xinyu Li:** Writing – original draft, Formal analysis, Data curation, Writing – review & editing. **Lin Zhu:** Conceptualization, Methodology, Writing – review & editing. **Hongfu Sun:** Conceptualization, Writing – review & editing. **Jun Li:** Methodology, Writing – review & editing. **Ximing Lv:** Data curation. **Chengli Qi:** Resources. **Huanhuan Yan:** Resources.

## Acknowledgments

This research was supported by a National Natural Science Foundation of China grant Nos. 12292983 and 42271383. And special thanks to Professor Di di from Nanjing University of Information Science and Technology for her advice and assistance on Jacobian calculations.

## Declaration of competing interest

The authors declare that they have no known competing financial interests or personal relationships that could have appeared to influence the work reported in this paper.

## Data availability

Atmosphere profile data are available via https://doi.org/10.5281/zenodo.14174378. TROPOMI $SO_2$ data are freely available via https://doi.org/10.5270/S5P-74eidii. The LBLRTM code are freely available via https://doi.org/10.5281/zenodo.3837549. The ERA5 specific humidity data are freely available from the Copernicus Climate Change Service (C3S) Climate Data Store (CDS; https://doi.org/10.24381/cds.adbb2d47).

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
