# Peer review of "A channel selection methodology for enhancing volcanic SO2 monitoring using FY-3E/HIRAS-II hyperspectral data"

_Atmospheric Measurement Techniques, 2024_

## Referee Comment (RC1)

General comments:

In this manuscript, the authors present a novel methodology on channel selection from FY-3E/HIRAS-II hyperspectral IR to detect $SO_2$ while eliminating the impact from temperature and moisture in the atmosphere. The topic is interesting and would be beneficial for future applications on $SO_2$ quantitative retrievals. However, there remains some questions that are not well clarified in the manuscript. My major comment is that the title of this research is kind of misleading as it says 'quantitatively monitor'. This would, to some extent, imply the retrieval of $SO_2$ levels from satellite observations which never show up in this research. This research is mainly focused on channel selection, but sadly it's not reflected in the title. Therefore, I would suggest the authors revise the title of manuscript to better reflect the key contents of the research, and go through a round of revision to address the specific comments before it is published.

Specific comments:

1. Line 40, the full name of 'UV' should be given here as it appears in the manuscript for the first time.

2. Line 49, polar orbiting hyperspectral sounders observe the same area in a period no less than 12 hours, which is not enough to be described as 'continuous observations'.

3. Line 65, the last segment is recommended to be revised as 'with both,,, and ,,, taken into consideration'.

4. Line 87, is that a typo of 'Radiative Transfer Model'?

5. Line 96, there's no T existing in equation (1), with only a $T_{sun}$ which is not 'true atmospheric temperature'.

6. Line 161, ERA5 has 37 fixed pressure levels vertically, and 137 model levels distributed using hybrid sigma-pressure coordinate system. It seems like you're using the model levels. It is recommended to point this out explicitly in the manuscript.

7. Figure 4, on the figure it seems like the selection of water vapor channels only depends on cross-comparison with temperature channels. According to lines 219 to 220, with the selected $SO_2$ channels being a subset that aligns with the water vapor channels (purple links), there should also be a cyan link between water vapor channels and $SO_2$ channels which points to water vapor selections. Or as illustrated in the figure, the relevant contents should be like 'the water vapor Jacobian of $SO_2$ channels must match those of the water vapor channel, while the temperature Jacobian of water vapor channels must match those of the $SO_2$ channels.'

8. Line 232, it seems like the additional $SO_2$ signal is around 1125 $cm^{-1}$ rather than 1225 $cm^{-1}$ from Figure 5.

9. There should be another set of figures between Figure 6 and 7 showing the temperature Jacobian functions of the channels within $SO_2$ absorption region.

10. Similar to comment #8, it seems like the left circle in Figure 5 is not around 1225 cm$^{-1}$, and not included in Figure 8.

11. Line 307, there should be a more detailed explanation on how a higher BT simulated with positive TD would indicate better $SO_2$ detection. Isn't it the variation of Jacobian that represent the detection ability better?

12. Line 348, the red box is on Figure 13(c) rather than 12(c).

---

## Author Comment (AC1)

**Replies to referees:**

We thank both referees for their careful reading of our manuscript. The comments helped us improve the paper. We provide a point-by-point reply to the comments below.

February 5, 2025

**Replies to Reviewer 2**

We thank the reviewer for the valuable comments and suggestions, which have improved the presentation of the paper.

★ *General Comments: In this work the Hyperspectral Infrared Atmospheric Sounder Type II (HIRAS-II), aboard the Fengyun 3E (FY-3E) satellite, is used to investigate the possibility to detect and retrieve the $SO_2$ emitted from volcanic eruptions. To do that, a methodology is described in order to select the most sensitive channels to $SO_2$ from the large number of hyperspectral channels recorded by the sensor. To minimize the influence of atmospheric water vapor and temperature to $SO_2$, the procedure proposes to select $SO_2$-sensitive channels that contain similar information on variations in atmospheric temperature and water vapor themself. Finally, to test the procedure, the 29 April 2024 eruption of Mount Ruang in Indonesia has been considered.*

*Here, the possibility of using FY-3E - HIRAS-II for monitoring eruptive volcanic clouds is shown. This polar sensor is part of the set of polar and geostationary satellites working at different wavelengths used for the $SO_2$ monitoring, whose synergic use can significantly improve the monitoring of these natural phenomena. The proposed procedure is interesting but needs to be clarified in several parts. The considered test case shows that there is a qualitative analogy between the $SO_2$ cloud detected by HIRAS-II and that detected by TROPOMI on board Sentinel 5p.*

● Response: We sincerely appreciate your thorough review of our manuscript and your valuable feedback. Your comments have played a crucial role in enhancing the scientific rigor and completeness of our work. Additionally, we are grateful for your encouragement of our research efforts.

In the revised manuscript, we reselected the absorption regions for $SO_2$ and water vapor based on their spectral absorption characteristics. Additionally, we determined appropriate $SO_2$ perturbation thresholds to ensure that the results more

accurately represent the gas distribution features in real volcanic eruption scenarios. Based on this, we obtained the final channel selection results. Furthermore, we conducted additional experiments to validate the sensitivity of the $SO_2$ channels and their suitability for volcanic $SO_2$ detection. In response to the issues you raised, we have provided detailed replies in the manuscript, and these revisions and additions are fully reflected in the updated version.

★ *Specific comments:*

1. *- r23: in this work only qualitative information are extracted.*

● Response: Thank you for your comments. In the revision, we have removed the description of "quantitative" in the manuscript and revised the content as follows:

Using FY-3E/HIRAS-II measurements, the spatial distribution and qualitative information of volcanic $SO_2$ are easily observed. (**Revised manuscript line 25**)

2. *- r94: clarify the reference, Li et al., 1994 doesn't contain the equation inserted. Check also the sign of the different terms and define theta.*

● Response: Thank you for pointing out this problem. In the revised manuscript, we have corrected the cited references and thoroughly checked the parameter symbols in the equations to ensure their accuracy. The corrected reference is as follows:

- Li, J.: Temperature and water vapor weighting functions from radiative transfer equation with surface emissivity and solar reflectivity. Adv. Atmos. Sci., 11, 421-426, doi:10.1007/BF02658162, 1994.

Additionally, we have supplemented the definitions of $T_{sun}$, $T_s$ and $\theta$ as they pertain to the equation for clarity, where $T_{sun}$ is solar temperature, $T_s$ is surface temperature and $\theta$ is the zenith angle. (**Revised manuscript lines 97-100**)

3. *- r96: T is not present in the formula. You could explicit the dipendence from T in the planck function (by written Bs(Ts) in the first term and B(T) into the integral.*

● Response: Thank you very much for your insightful comments. In the revised manuscript, we have explicitly indicated the dependence of T and B within the

Planck function in the equation (1) accompanied by appropriate annotations and explanations:

$$R = \varepsilon B_s(T_s)\tau_s - \int_0^{P_s} B(T)d\tau + (1-\varepsilon)\int_0^{P_s} B(T)d\tau^* + 2.16 \times 10^{-5}\pi\cos\theta \times$$

$$\rho_r B_r(T_{sun}) \times {\tau_s}^2$$

(**Revised manuscript line 97**)

4. *- r109-110: clarify if LBLRTM allows the possibility to insert a user defined atmospheric PTH profiles.*

● Response: Thank you for your suggestions. In the revised manuscript, we have clarified that the LBLRTM allows users to customize input profile files. Additionally, we have specified the content of the profiles used in our study. The details are as follows:

LBLRTM allows for the input of user-defined atmospheric profile files. In this study, the meteorological data input into LBLRTM consists of six standard atmospheric profiles: the US Standard Atmosphere, 1976, and profiles for mid-latitude summer, mid-latitude winter, subarctic summer and subarctic winter. (**Revised manuscript line 113**)

5. *- r116 Paragraph 2.2: are the HIRAS-II data freely available? Where they can be downloaded? This information could be inserted in the text.*

● Response: Thank you for your valuable feedback. FY-3E/HIRAS-II data are freely available from the FENGYUN Satellite Data Service (https://satellite.nsmc.org.cn/DataPortal/cn/home/index.html). We have incorporated these contents into the revised manuscript. (**Revised manuscript line 133**)

6. *- r125-r127: what about the NEdT for the short-wave bands?*

● Response: Thank you for raising this question. Since our study did not utilize spectral channels in the shortwave band, we did not initially include relevant information on this band. However, based on your suggestion, we realized that adding the NEDT information for the shortwave band would enhance the overall

coherence of the manuscript and provide a more comprehensive description of the data. In the revised manuscript, we have incorporated this information as follows:

Based on the radiometric specifications for FY-3E/HIRAS-II, the noise equivalent target brightness temperature (BT) difference (NEdT) is specified within 0.2 – 0.4 K for the long-wave IR band, 0.2 – 0.3 K (at 280 K) for the mid-wave IR band and 0.8 – 2.4 K (at 280 K) for the short-wave IR band (Huang et al., 2023). (**Revised manuscript line 131**)

7. *- Table 1: as written in the paper "Its measurements span a continuous spectrum range of 648.75 to 2551.25 cm⁻¹ at a resolution of 0.625 cm⁻¹". In the table seems that the different spectral intervals are not in continuity. For example: the Long spectral range ended at 1136 cm⁻¹ and the Mid spectral range start at 1210 cm cm⁻¹ (lack of 74 cm⁻¹). Why some channels have been not considered?*

● Response: Thank you for your questions. We sincerely thank the reviewer for their careful observation regarding the spectral range distribution in our study. After reviewing the relevant literature and official technical documentation of FY-3E/HIRAS-II, we found that the HIRAS L1 data are released in two spectral resolution modes: Full Resolution (FR) and Design Resolution (DR) (Li et al., 2022). In the originally submitted manuscript, we mistakenly presented the spectral range distribution of different resolution modes as the Full Resolution data, and we apologize for this oversight. In fact, our study utilizes the Full Resolution data from FY-3E/HIRAS-II, with a spectral resolution of 0.625 cm⁻¹. In the revised manuscript, we have corrected this error and updated the relevant tables and data descriptions to ensure the accuracy and consistency of our results. (**Revised manuscript Table 1**)

Table 1 Spectral parameters of FY-3E/HIRAS-II channels (Xie et al., 2023)

| IR Wave Band | Spectral Range (cm⁻¹) | No. of Channels | Spectral Resolution (cm⁻¹) |
|---|---|---|---|
| Long | 648.75 – 1169.375 | 834 | 0.625 |

| | (15.41 – 8.55 μm) | | |
|---|---|---|---|
| Mid | 1167.5 – 1921.25 (8.56 – 5.20 μm) | 1207 | 0.625 |
| Short | 1919.375 – 2551.25 (5.21 – 3.92 μm) | 1012 | 0.625 |

8. *- r139: citation not present in the bibliography.*

● Response: Thank you for your suggestion. In the revised manuscript, we have added the relevant reference to the reference list. The added reference is as follows:

- Corradino, C., Jouve, P., La Spina, A.Del Negro, C.: Monitoring Earth's atmosphere with Sentinel-5 TROPOMI and Artificial Intelligence: Quantifying volcanic $SO_2$ emissions. Remote Sensing of Environment, 315, 114463, doi:https://doi.org/10.1016/j.rse.2024.114463, 2024.

    (**Revised manuscript line 509**)

9. *- r141-r142: Please insert the references:*

*- Theys, N.; De Smedt, I.; Yu, H.; Danckaert, T.; Van Gent, J.; Hörmann, C.; Wagner, T.; Hedelt, P.; Bauer, H.; Romahn, F.; et al. Sulfur dioxide retrievals from TROPOMI onboard Sentinel-5 Precursor: Algorithm theoretical basis. Atmos. Meas. Tech. 2017, 10, 119−153.*

*- Theys, N.; Hedelt, P.; De Smedt, I.; Lerot, C.; Yu, H.; Vlietinck, J.; Pedergnana, M.; Arellano, S.; Galle, B.; Fernandez, D.; et al. Global monitoring of volcanic $SO_2$ degassing with unprecedented resolution from TROPOMI onboard Sentinel-5 Precursor. Sci. Rep. 2019, 9, 1−10.*

● Response: Thank you for your suggestions. We have carefully reviewed the two references and found that the content provides significant insights and valuable guidance for our work. We have included these references in the revised manuscript, as detailed below:

    Daily or sub-daily revisits of specific sites are achievable, given TROPOMI's 108° cross-orbit field of view and its ability to capture data across multiple orbits

(Theys et al., 2017). Since 2019, Sentinel-5P's spatial resolution has been enhanced to 3.5 km × 5.5 km. TROPOMI measures data across four spectral regions (ultraviolet, visible, near-infrared, and shortwave infrared) and is adept at monitoring $SO_2$ and a range of other gases (Theys et al., 2019). (**Revised manuscript line 147-149**)

10. *- Figure 3: enlarge the x and y number labels (as in plot (b)). You should use ppbv instead of ppmv (in both plots). Here the brightness temperature is indicated as $T_b$, while in the text with BT, please standardize.*

● Response: Thank you for pointing out this problem. In the revised manuscript, we have redrawn Figure 3. Based on your suggestion, we have made the following modifications in the new image:

**First**, we have increased the font size of the x and y labels and ensured consistency between the two subfigures.

**Second**, we have changed the sulfur dioxide concentration to ppbv and applied this modification throughout the study.

**Finally**, we have updated the brightness temperature symbol in the image to BT, ensuring consistency with the rest of the manuscript. (**Revised manuscript Sec. 3.2 Figure 3**)

[Figure]

Figure 3: Representation of the maximum half-width and peak value of the SO₂ Jacobian function for the US Standard Atmosphere, 1976: (a) SO₂ profile, (b) 1163.125 cm⁻¹ channel.

11. *- r217-r219: explain better why the similarity in the Jacobians in the different spectral ranges is important to minimize the influence of water vapour and temperature to SO₂.*

● Response: Thank you for raising this important question. As suggested, we have enhanced the explanation in the revised manuscript regarding the significance of Jacobian similarity across spectral ranges in mitigating water vapor ($H_2O$) and temperature (T) interference on SO₂ retrievals. The rationale can be summarized in two key aspects:

**(1) Radiance sensitivity to the atmospheric information**

CO₂ absorption channels primarily reflect the information of atmospheric temperature profiles (Li et al., 2022), water vapor channels contain both temperature and water vapor information, while $SO_2$ channels contain information of temperature, water vapor and $SO_2$. To separate temperature from water vapor in water vapor absorption channel radiances, $CO_2$ channels play an important role through providing temperature information. If a water vapor absorption channel

and a $CO_2$ absorption channel have similar temperature Jacobian, they have also similar temperature sensitivity, and thus that $CO_2$ channel is helpful for separating the temperature from water vapor in the water vapor channel radiance. Same for a $SO_2$ channel, if a water vapor channel has similar temperature Jacobian and water vapor Jacobian, then the water vapor channel is helpful for separating temperature and water vapor from $SO_2$ in that $SO_2$ channel radiance.

**(2) Jacobian Consistency Protocol**

Through inter-channel Jacobian matching, we ensure that the variations in water vapor Jacobian matrix and temperature Jacobian matrix within the water vapor absorption region are consistent with those in the $SO_2$ channels.

Thus, when subtracting the brightness temperature of the $SO_2$ channel from that of the water vapor channel, the influence of water vapor, atmospheric temperature, and surface radiation shared by both channels can be effectively.

**This methodology effectively decouples $SO_2$ signals from confounding atmospheric states, with full implementation details provided in the revised manuscript (Lines 229-244).**

12. *- Figure 4: This scheme it is not so clear to me:*

*is it correct that the spectral range selected for the water vapor absorption region is the same as for the $SO_2$ absorption region? In this case the water vapor Jacobian marix (computed for a specific wavenumber, by varying the water vapour content) should be the same. The water vapor selection is in this case carried out by considering the maximum M and dP? In the scheme seems that only the cross-comparison between the water vapour Jacobians lead to the selection of the $SO_2$ channels. Is it correct? Moreover, it is not also clear to me why only the 1155-1430 interval is considered for the $SO_2$ Jacobian computation. $SO_2$ presents two wide absorption bands around 1163 and 1370 $cm^{-1}$, and until 1100 $cm^{-1}$ the $SO_2$ absorption is still meaningful. Why the whole 1100-1430 $cm^{-1}$ spectral range it is not considered? I'm surprise to see that no one channels around 1163 $cm^{-1}$ is selected. This $SO_2$ absorption is inside of one of the TIR atmospheric window and*

*generally used for the SO$_2$ tropospheric retrievals.*

● Response: Thank you very much for your valuable suggestions and comments. We will address your comments from two aspects:

**(1)** We recognized that the previously selected water vapor absorption region had certain limitations. Therefore, based on the spectral absorption characteristics of water vapor, we reselected the water vapor absorption region within the range of 1400 – 1920 cm$^{-1}$ (7.14 – 5.20 μm) and re-screened the water vapor channels within this new absorption region (Rodimova, 2018). Additionally, we redrew Figure 4 to more clearly illustrate the channel selection process.

Specifically, the SO$_2$ channels were selected based on the Jacobian information analysis method. The atmospheric temperature channels were determined by comparing the temperature Jacobians in the CO$_2$ absorption region with those of the selected SO$_2$ channels. The selection of water vapor channels was conducted in two steps: first, by comparing the temperature Jacobians in the water vapor absorption region with those of the SO$_2$ channels; second, by comparing the water vapor Jacobians in the water vapor absorption region with those of the SO$_2$ channels. Ultimately, we identified suitable water vapor channels that have both similar temperature and water vapor Jacobians of SO$_2$ channels. Note that we identified SO$_2$ channels first, then found water vapor channels with similar Jacobians, those selected water vapor channels do not have SO$_2$ absorption, meaning there is no overlapping channel between selected water vapor channels and the SO$_2$ channels. The idea on selecting CO$_2$ and water vapor channels with similar T/q Jacobians of SO$_2$ channels is to separate temperature and water vapor from SO$_2$ in the SO$_2$ channels radiances.

**(2)** We agree with the reviewer's comments regarding the selection of SO$_2$ channels and have revised our SO$_2$ channel selection accordingly. In the new scheme, we have expanded the SO$_2$ absorption region to 1100 – 1430 cm$^{-1}$. Since the 1100 – 1170 cm$^{-1}$ spectral region is highly effective for detecting SO$_2$ plumes in the troposphere and is particularly valuable for monitoring volcanoes characterized by continuous passive degassing (Carboni et al., 2016), we have

reselected SO₂ channels within this range. Given that the chosen channels can effectively capture SO₂ across different atmospheric layers, we have carefully selected channels from both $1100 - 1170$ and $1320 - 1370$ cm$^{-1}$ bands to ensure comprehensive coverage.

In conclusion, we have incorporated the necessary revisions into the revised manuscript, and provided Figure 4 and Figure 5 below. (**Revised manuscript Sec. 3.2**)

[Figure]

**Figure 4: Schematic diagram of channel selection method.**

[Figure]

**Figure 5: Schematic diagram of the SO₂ Jacobian matrix with atmospheric profiles from the US Standard Atmosphere, 1976.**

13. *- r226: the SO₂ perturbation is generated by varying a default profile of 5%. But,*

*during volcanic emission, the SO₂ content is much higher and also confined at specific layers. How the SO₂ Jacobian computed can be considered representative of a real case?*

- Response: Thank you for pointing out this problem. After reviewing relevant literature, we found that the background atmospheric $SO_2$ concentration is typically ranges from 0.25 – 0.43 ppbv. During volcanic eruptions, however, $SO_2$ concentrations are significantly higher, ranging from approximately 500 – 1000 ppbv. Therefore, we increased the $SO_2$ perturbation magnitude to $5 \times 10^4$ times the background $SO_2$ concentration in ppbv to ensure that our perturbation levels are representative of actual volcanic eruption scenarios.

  Additionally, although $SO_2$ in real volcanic eruption events is generally concentrated in specific atmospheric layers, volcanic eruptions are typically rapid and prolonged. To ensure the completeness of our simulation results, we applied perturbations to all 99 atmospheric layers from 0 to 1040 hPa in our study. The modified content in the manuscript is as follows :

  In situ measurements reported by Rose et al. (2004) indicate $SO_2$ concentrations of 500 – 1000 ppbv during an aircraft encounter with a 35-hour-old volcanic plume from the Icelandic Hekla eruption in February 2000, at a distance of approximately 1300 km from the source. In comparison, the concentration of $SO_2$ in the clean troposphere typically ranges from 0.25 – 0.43 ppbv (Casadevall et al., 1984). Given that $SO_2$ concentrations increase dramatically over a short period during volcanic eruptions, for $SO_2$, we perturb the atmospheric profiles at different pressure levels using $5 \times 10^4$ times gas content, to better represent the gas distribution characteristics in volcanic eruption scenarios. (**Revised manuscript line 251-256**)

14. *- r228-r231: except for the wavenumbers around 1360 cm⁻¹, Figure 5 doesn't clearly emphasize where are placed the other wavenumbers significant. In any case the left orange oval (that should emphasize the higher jacobian variability) is placed around 1210 cm⁻¹ and not 1225 cm⁻¹.*

- Response: Thank you for your valuable suggestions. We fully acknowledge the importance of channels within the 1100 – 1170 cm⁻¹ spectral range for $SO_2$ monitoring. Although this range is located within the atmospheric window and the $SO_2$ signal intensity is weaker compared to the 1320 – 1370 cm⁻¹ range, its significant advantage lies in its minimal susceptibility to water vapor interference. This characteristic makes the 1100 – 1170 cm⁻¹ range particularly valuable for monitoring $SO_2$ in the middle and lower troposphere. In the revised manuscript, we have further emphasized the importance of the 1100 – 1170 cm⁻¹ range and selected specific channels within this range for $SO_2$ monitoring, thereby enhancing the accuracy and reliability of $SO_2$ detection in the middle and lower troposphere. The revised content in the manuscript is as follows:

The 1360 cm⁻¹ band exhibits the strongest $SO_2$ signal among the available spectral bands. However, it is also a strong absorption region for atmospheric water vapor, which can introduce contamination in $SO_2$ retrievals. This band demonstrates minimal sensitivity to radiative contributions from the surface and lower atmosphere, making it particularly effective for monitoring stratospheric $SO_2$ plumes (Thomas & Watson, 2010). In contrast, the 1163 cm⁻¹ band falls within an atmospheric window region. While the presence of $SO_2$ in this band leads to a certain degree of radiative attenuation, it remains well-suited for detecting $SO_2$ plumes in the troposphere (Carboni et al., 2016). This characteristic makes it especially valuable for monitoring volcanic activity characterized by continuous passive degassing. By leveraging the complementary strengths of these bands, we select $SO_2$-sensitive channels with a central wavenumber around 1163 and 1360 cm⁻¹. (**Revised manuscript line 259-268**)

15. *- r306: it is not clear which channels have been selected, please clarify.*

- Response: Thank you for your comments. After reassessing the selection of $SO_2$ channels, we have ultimately identified two $SO_2$ channels, 1163.125 and 1360.625 cm⁻¹. Based on this, we have supplemented the sensitivity analysis of the 1163.125 cm⁻¹ channel with respect to the temperature difference between the surface and

near-surface air in our original study. In the revised manuscript, we have clearly specified the channels used for sensitivity analysis. Additionally, we have provided a detailed comparison and discussion of the results obtained from both the 1163.125 and 1360.625 $cm^{-1}$ channels. (**Revised manuscript Sec. 4.1**)

16. *- r317: why the 1165.125 $cm^{-1}$ channel has been considered? The Paragraph 3.2.1 (SO$_2$ channel selection) indicates only the channels around 1360 $cm^{-1}$.*

● Response: Thank you for your question. In previous research, we conducted a sensitivity analysis of SO$_2$ plumes using the 1360.625 $cm^{-1}$ channel and observed significant signal saturation in this channel when SO$_2$ plume concentrations exceeded 200 DU. Simultaneously, we noted that the 1160.125 $cm^{-1}$ channel also contains partial SO$_2$ information. Based on this, we further analyzed the sensitivity of the 1163.125 $cm^{-1}$ channel to SO$_2$ plumes. The results demonstrated that the 1160.125 $cm^{-1}$ channel effectively avoids the saturation issues encountered in the 1360.625 $cm^{-1}$ channel under high SO$_2$ plume concentrations. Therefore, in the revised manuscript, we have selected both the 1163.125 and 1360.625 $cm^{-1}$ channels as the primary channels for SO$_2$ monitoring. This approach enhances the completeness of SO$_2$ channels selection, improving the accuracy and reliability of detection. (**Revised manuscript Sec. 3.2.1**)

**References:**

Carboni, E., Grainger, R.G., Mather, T.A., Pyle, D.M., Thomas, G.E., Siddans, R., et al.: The vertical distribution of volcanic $SO_2$ plumes measured by IASI. Atmos. Chem. Phys., 16, 4343-4367, doi:10.5194/acp-16-4343-2016, 2016.

Casadevall, T.J., Rose Jr., W.I., Fuller, W.H., Hunt, W.H., Hart, M.A., Moyers, J.L., et al.: Sulfur dioxide and particles in quiescent volcanic plumes from Poás, Arenal, and Colima Volcanos, Costa Rica and Mexico, J. Geophys. Res., 89, 9633-9641, doi:https://doi.org/10.1029/JD089iD06p09633, 1984.

Huang, J., Ma, G., Liu, G.Q., Li, J.Zhang, H.: The Evaluation of FY-3E Hyperspectral Infrared Atmospheric Sounder-II Long-Wave Temperature Sounding Channels, Remote Sens., 15, 17, doi:10.3390/rs15235525, 2023.

Li, S., Hu, H., Fang, C., Wang, S., Xun, S., He, B., et al.: Hyperspectral Infrared Atmospheric Sounder (HIRAS) Atmospheric Sounding System. 14, 3882, 2022.

Rodimova, O.B.: Carbon Dioxide and Water Vapor Continuum Absorption in the Infrared Spectral Region. Atmospheric and Oceanic Optics, 31, 564-569, doi:10.1134/S1024856018060143, 2018.

Rose, W.I., Gu, Y., Watson, I.M., Yu, T., Blut, G.J.S., Prata, A.J., et al.: The February–March 2000 Eruption of Hekla, Iceland from a Satellite Perspective, Volcanism and the Earth's Atmosphere, pp. 107-132, 2004.

Theys, N., De Smedt, I., Yu, H., Danckaert, T., van Gent, J., Hörmann, C., et al.: Sulfur dioxide retrievals from TROPOMI onboard Sentinel-5 Precursor: algorithm theoretical basis. Atmos. Meas. Tech., 10, 119-153, doi:10.5194/amt-10-119-2017, 2017.

Theys, N., Hedelt, P., De Smedt, I., Lerot, C., Yu, H., Vlietinck, J., et al.: Global monitoring of volcanic $SO_2$ degassing with unprecedented resolution from TROPOMI onboard Sentinel-5 Precursor. Scientific Reports, 9, 2643, doi:10.1038/s41598-019-39279-y, 2019.

Thomas, H.E.Watson, I.M.: Observations of volcanic emissions from space: current and future perspectives. Natural Hazards, 54, 323-354, doi:10.1007/s11069-009-9471-3, 2010.

Xie, M., Gu, M., Hu, Y., Huang, P., Zhang, C., Yang, T., et al.: A Study on the Retrieval of Ozone Profiles Using FY-3D/HIRAS Infrared Hyperspectral Data, Remote Sens., 15, 1009, doi: https://doi.org/10.3390/rs15041009, 2023.

---

## Author Comment (AC2)

**Replies to referees:**

We thank both referees for their careful reading of our manuscript. The comments helped us improve the paper. We provide a point-by-point reply to the comments below.

February 5, 2025

**Replies to Reviewer 1**

We thank the reviewer for the valuable comments and suggestions, which have improved the presentation of the paper.

★ *General comments: In this manuscript, the authors present a novel methodology on channel selection from FY-3E/HIRAS-II hyperspectral IR to detect SO₂ while eliminating the impact from temperature and moisture in the atmosphere. The topic is interesting and would be beneficial for future applications on SO₂ quantitative retrievals. However, there remains some questions that are not well clarified in the manuscript. My major comment is that the title of this research is kind of misleading as it says 'quantitatively monitor'. This would, to some extent, imply the retrieval of SO₂ levels from satellite observations which never show up in this research. This research is mainly focused on channel selection, but sadly it's not reflected in the title. Therefore, I would suggest the authors revise the title of manuscript to better reflect the key contents of the research, and go through a round of revision to address the specific comments before it is published.*

● Response: Thank you for your careful review and valuable comments on our paper. In this revision, we modified the manuscript title to better reflect the focus of our study on channel selection, making the title clearer and more precise. We have revised the manuscript title to:

A channel selection methodology for enhancing volcanic SO₂ monitoring using FY-3E/HIRAS-II hyperspectral data

In the revised manuscript, we reselected the absorption regions for SO₂ and water vapor based on their spectral absorption characteristics. Additionally, we determined appropriate SO₂ perturbation thresholds to ensure that the results more accurately represent the gas distribution features in real volcanic eruption scenarios. Based on this, we obtained the final channel selection results. Furthermore, we

conducted additional experiments to validate the sensitivity of the SO₂ channels and their suitability for volcanic SO₂ detection. In response to the issues you raised, we have provided detailed replies in the manuscript, and these revisions and additions are fully reflected in the updated version.

★ *Specific comments:*

1. *Line 40, the full name of 'UV' should be given here as it appears in the manuscript for the first time.*

● Response: Thank you very much for your comments. In the revision, we included the full name of UV, 'Ultraviolet,' in the manuscript to provide greater clarity in the content. The revised content in the manuscript is as follows:

   Ultraviolet (UV) band sensors are limited to monitoring SO₂ from daytime eruptions due to their reflective nature. (**Revised manuscript line 42**)

2. *Line 49, polar orbiting hyperspectral sounders observe the same area in a period no less than 12 hours, which is not enough to be described as 'continuous observations'.*

● Response: Thank you very much for your suggestions. We agree with the reviewer's opinion and have modified the term 'continuous observation' in the manuscript to improve the rigor and reliability of the content. The explanation is as follows:

   Hyperspectral IR sensors enable observations with finer channel bandwidths that accurately characterize and distinguish each component, thereby reducing interference from other materials. (**Revised manuscript line 51**)

3. *Line 65, the last segment is recommended to be revised as 'with both,,, and ,,, taken into consideration'.*

● Response: Thank you very much for your comments. In accordance with your suggestions, to ensure that the language aligns more closely with academic standards, we have modified the original text as:

   Lipton (2003) developed a method to select atmospheric microwave sounding

channels based on the combination of each channel's center frequency, bandwidth, and degrees of freedom for the signal, with both applicability to multiple environmental conditions and providing robust retrieval performance taken into consideration. (**Revised manuscript line 67**)

4. *Line 87, is that a typo of 'Radiative Transfer Model'?*

● Response: We sincerely appreciate your meticulous attention in identifying this typographical error, and we extend our apologies for this oversight. In the revised manuscript, we have rectified the error in question and conducted a thorough review of the entire document to preclude the occurrence of similar inaccuracies. (**Revised manuscript line 90**)

5. *Line 96, there's no T existing in equation (1), with only a $T_{sun}$ which is no 'true atmospheric temperature'.*

● Response: Thank you very much for your insightful comments. In the revised manuscript, we have explicitly indicated the dependence of T and B within the Planck function in the equation (1) accompanied by appropriate annotations and explanations:

$$R = \varepsilon B_s(T_s)\tau_s - \int_0^{P_s} B(T)d\tau + (1-\varepsilon)\int_0^{P_s} B(T)d\tau^* + 2.16 \times 10^{-5}\pi \cos\theta \times$$
$$\rho_r B_r(T_{sun}) \times \tau_s^2$$

Additionally, we have supplemented the definitions of $T_{sun}$, $T_s$ and $\theta$ as they pertain to the equation for clarity, where $T_{sun}$ is solar temperature, $T_s$ is surface temperature and $\theta$ is the zenith angle. (**Revised manuscript lines 97-100**)

6. *Line 161, ERA5 has 37 fixed pressure levels vertically, and 137 model levels distributed using hybrid sigma-pressure coordinate system. It seems like you're using the model levels. It is recommended to point this out explicitly in the manuscript.*

● Response: Thank you for your suggestions. We utilized ERA5 hourly specific humidity data on pressure levels from 1940 to present at the 400 hPa level from

the 37 fixed pressure levels of ERA5. This data was used to validate that our selected channels combination effectively removes the influence of water vapor interference in sulfur dioxide monitoring. The ERA5 data used in this study can be accessed via the Copernicus Climate Change Service (C3S) Climate Data Store (CDS; https://doi.org/10.24381/cds.adbb2d47). In this revision, we have provided a detailed description of the data types utilized as follows to give a clearer understanding:

Each profile from ERA5 has a horizontal scale of 31 km. and This includes upper-air parameters on 37 fixed pressure levels from 1,000 to 1 hPa and 137 model levels distributed using hybrid sigma-pressure coordinate system.137 vertical levels, ranging from near-surface air pressure to 0.01 hPa. For this study, we interpolate ERA5 400 hPa fixed pressure level data to assess atmospheric water vapor conditions near Mount Ruang concurrent with FY-3E/HIRAS-II observations. (**Revised manuscript line 167**)

7. *Figure 4, on the figure it seems like the selection of water vapor channels only depends on cross-comparison with temperature channels. According to lines 219 to 220, with the selected $SO_2$ channels being a subset that aligns with the water vapor channels (purple links), there should also be a cyan link between water vapor channels and $SO_2$ channels which points to water vapor selections. Or as illustrated in the figure, the relevant contents should be like 'the water vapor Jacobian of $SO_2$ channels must match those of the water vapor channel, while the temperature Jacobian of water vapor channels must match those of the $SO_2$ channels.'*

- Response: Thank you for your feedback. We fully concur with the reviewer's observations and have comprehensively revised Figure 4 to address these comments. In the updated figure:

  1. $SO_2$ channel selection was guided by Jacobian analysis.

  2. Atmospheric temperature channels were determined through comparative analysis of temperature Jacobians between the atmospheric temperature absorption region and the pre-selected $SO_2$ channels.

3. Water vapor channel selection employed a two-stage process:

First, temperature Jacobians from the water vapor absorption region were cross-compared with those from $SO_2$ channels

Second, water vapor Jacobians from the same region were analyzed against corresponding Jacobians from $SO_2$ channels.

This systematic approach yielded optimal water vapor channel selections. Note that the selected water vapor channels with similar temperature and water vapor Jacobians of $SO_2$ channels do not contain $SO_2$ absorption, meaning there is no overlapping channel between selected water vapor channels and $SO_2$ channels. **(Revised manuscript Section 3.2, Figure 4)**.

[Figure]

**Figure 4: Schematic diagram of channel selection method.**

8. *Line 232, it seems like the additional $SO_2$ signal is around 1125 cm$^{-1}$ rather than 1225 cm$^{-1}$ from Figure 5.*

● Response: Thank you for pointing out the problem. To more comprehensively characterize the absorption capacity of satellite channels for $SO_2$, we have expanded the $SO_2$ absorption range in our study to $1100 – 1430$ cm$^{-1}$. Additionally, although the $SO_2$ signal in the $1100 – 1170$ cm$^{-1}$ range is relatively weak, it remains significant for monitoring tropospheric $SO_2$. Therefore, we have also included $SO_2$ monitoring channels within the $1100 – 1170$ cm$^{-1}$ range to achieve more accurate $SO_2$ detection. **(Revised manuscript Sec. 3.2.1, Figure 5)**

[Figure]

**Figure 5: Schematic diagram of the SO₂ Jacobian matrix with atmospheric profiles from the US Standard Atmosphere, 1976.**

9. *There should be another set of figures between Figure 6 and 7 showing the temperature Jacobian functions of the channels within SO₂ absorption region.*

● Response: Thank you very much for the constructive feedback. We have redefined the spectral ranges for the SO₂ absorption region and the water vapor absorption region. Following your suggestion, we have supplemented the sections on atmospheric temperature channel selection and water vapor channel selection with Jacobian figures for atmospheric temperature and water vapor within the SO₂ absorption region, respectively. In these figures, we have annotated the SO₂ channels selected in the study to clearly indicate their Jacobian peak values. The following are the temperature Jacobian figures and water vapor Jacobian figures for the SO₂ absorption region. (**Revised manuscript Sec. 3.2.2 and Sec. 3.2.3**)

[Figure]

**Figure 7: Representations of temperature Jacobian functions at SO₂ absorption region (black dashed lines represent selected SO₂ channels) for the conditions of six atmospheric profiles: (a) tropical atmospheric profile, (b) mid-latitude summer atmospheric profile, (c) mid-latitude winter atmospheric profile, (d) subarctic summer atmospheric profile, (e) subarctic winter atmospheric profile, and (f) US Standard Atmosphere, 1976.**

[Figure]

**Figure 9: Representations of water vapor Jacobian functions at SO₂ absorption region (black dashed lines represent selected SO₂ channels) for conditions of six atmospheric profiles: (a) tropical atmospheric profile, (b) mid-latitude summer atmospheric profile, (c) mid-latitude winter atmospheric profile, (d) subarctic summer atmospheric profile, (e) subarctic winter atmospheric profile, and (f) US Standard Atmosphere, 1976.**

10. *Similar to comment #8, it seems like the left circle in Figure 5 is not around 1225 cm⁻¹, and not included in Figure 8.*

● Response: Thank you for pointing out this problem. We apologize for this oversight. In the revised manuscript, we have expanded the sulfur dioxide absorption

spectrum range and selected sulfur dioxide channels in the $1100 - 1170 \text{ cm}^{-1}$ spectral range to obtain more comprehensive sulfur dioxide information. Figure 5 has already been presented in the response to comment 8. (**Revised manuscript Sec. 3.2.1**)

11. *Line 307, there should be a more detailed explanation on how a higher BT simulated with positive TD would indicate better SO₂ detection. Isn't it the variation of Jacobian that represent the detection ability better?*

- Response: Thank you for your questions. For a specified wavenumber (v), the sensitivity of BT to variations in geophysical parameters (X) is represented by the Jacobian matrix for each pressure layer as follows: $J_v(X) = \frac{\partial BT(v)}{\partial X}$. The Jacobian formula defines the relationship between the change in brightness temperature and the perturbation in material concentration. Under consistent atmospheric conditions with fixed SO₂ concentration perturbations and uniform background brightness temperature, the brightness temperature after SO₂ perturbation demonstrates a trend and relative behavior similar to that of the Jacobian value. As a result, brightness temperature can effectively substitute for the Jacobian value in assessing the detection capability of SO₂. Based on your suggestion, we have added a more detailed explanation in the revised manuscript on how a higher BT simulated with positive TD would indicate better SO₂ detection. The modified content in the manuscript is as follows:

    The Jacobian formula defines the relationship between the change in brightness temperature and the perturbation in material concentration. Under consistent atmospheric conditions with fixed SO₂ concentration perturbations and uniform background brightness temperature, the TD after SO₂ perturbation demonstrates a similar trend and behavior to that of the Jacobian value. As a result, TD can effectively substitute for the Jacobian value in assessing the detection capability of SO₂. (**Revised manuscript line 342**)

12. *Line 348, the red box is on Figure 13(c) rather than 12(c).*

● Response: We appreciate your attention to this problem. We have made the necessary correction in the revised manuscript, and the red box is now correctly positioned in Figure 14(b). (**Revised manuscript Sec. 5 Figure 14**)

[Figure]

**Figure 14: FY-3E/HIRAS-II brightness temperature difference data for the region around Mount Ruang (black star in each image) at 08:55 UT on 18 April 2024 for the channels (a) 1360.625 and 902.5 cm$^{-1}$, and (b) 1360.625 and 1429.375 cm$^{-1}$, (c) 1163.125 and 902.5 cm$^{-1}$ and (d) 1163.125 and 1887.5 cm$^{-1}$.**

---

## Author Response (AR1)

**Replies to referees:**

We thank both referees for their careful reading of our manuscript. The comments helped us improve the paper. We provide a point-by-point reply to the comments below.

February 5, 2025

**Replies to Reviewer 1**

We thank the reviewer for the valuable comments and suggestions, which have improved the presentation of the paper.

- ★ General comments: In this manuscript, the authors present a novel methodology on channel selection from FY-3E/HIRAS-II hyperspectral IR to detect SO₂ while eliminating the impact from temperature and moisture in the atmosphere. The topic is interesting and would be beneficial for future applications on SO₂ quantitative retrievals. However, there remains some questions that are not well clarified in the manuscript. My major comment is that the title of this research is kind of misleading as it says 'quantitatively monitor'. This would, to some extent, imply the retrieval of SO₂ levels from satellite observations which never show up in this research. This research is mainly focused on channel selection, but sadly it's not reflected in the title. Therefore, I would suggest the authors revise the title of manuscript to better reflect the key contents of the research, and go through a round of revision to address the specific comments before it is published.
- Response: Thank you for your careful review and valuable comments on our paper. In this revision, we modified the manuscript title to better reflect the focus of our study on channel selection, making the title clearer and more precise. We have revised the manuscript title to:

A channel selection methodology for enhancing volcanic SO2 monitoring using FY-3E/HIRAS-II hyperspectral data

In the revised manuscript, we reselected the absorption regions for  $SO_2$  and water vapor based on their spectral absorption characteristics. Additionally, we determined appropriate  $SO_2$  perturbation thresholds to ensure that the results more accurately represent the gas distribution features in real volcanic eruption scenarios. Based on this, we obtained the final channel selection results. Furthermore, we conducted additional experiments to validate the sensitivity of the SO2 channels and their suitability for volcanic SO2 detection. In response to the issues you raised, we have provided detailed replies in the manuscript, and these revisions and additions are fully reflected in the updated version.

**★ Specific comments:**

- 1. Line 40, the full name of 'UV' should be given here as it appears in the manuscript for the first time.
- Response: Thank you very much for your comments. In the revision, we included the full name of UV, 'Ultraviolet,' in the manuscript to provide greater clarity in the content. The revised content in the manuscript is as follows:

Ultraviolet (UV) band sensors are limited to monitoring SO2 from daytime eruptions due to their reflective nature. (**Revised manuscript line 42**)

- 2. Line 49, polar orbiting hyperspectral sounders observe the same area in a period no less than 12 hours, which is not enough to be described as 'continuous observations'.
- Response: Thank you very much for your suggestions. We agree with the reviewer' s opinion and have modified the term 'continuous observation' in the manuscript to improve the rigor and reliability of the content. The explanation is as follows:

Hyperspectral IR sensors enable observations with finer channel bandwidths that accurately characterize and distinguish each component, thereby reducing interference from other materials. (**Revised manuscript line 51**)

- 3. *Line 65, the last segment is recommended to be revised as 'with both,,, and ,,, taken into consideration'.*
- Response: Thank you very much for your comments. In accordance with your suggestions, to ensure that the language aligns more closely with academic standards, we have modified the original text as:

Lipton (2003) developed a method to select atmospheric microwave sounding

channels based on the combination of each channel's center frequency, bandwidth, and degrees of freedom for the signal, with both applicability to multiple environmental conditions and providing robust retrieval performance taken into consideration. (**Revised manuscript line 67**)

- 4. Line 87, is that a typo of 'Radiative Transfer Model'?
- Response: We sincerely appreciate your meticulous attention in identifying this typographical error, and we extend our apologies for this oversight. In the revised manuscript, we have rectified the error in question and conducted a thorough review of the entire document to preclude the occurrence of similar inaccuracies. (Revised manuscript line 90)
- 5. Line 96, there's no T existing in equation (1), with only a  $T_{sun}$  which is no 'true atmospheric temperature'.
- Response: Thank you very much for your insightful comments. In the revised manuscript, we have explicitly indicated the dependence of T and B within the Planck function in the equation (1) accompanied by appropriate annotations and explanations:

$$R = \varepsilon B_s(T_s)\tau_s - \int_0^{P_s} B(T)d\tau + (1-\varepsilon)\int_0^{P_s} B(T)d\tau^* + 2.16 \times 10^{-5}\pi\cos\theta \times 10^{-5}\pi\cos\theta$$

 $\rho_r B_r(T_{sun}) \times \tau_s^2$

Additionally, we have supplemented the definitions of  $T_{sun}$ ,  $T_s$  and  $\theta$  as they pertain to the equation for clarity, where  $T_{sun}$  is solar temperature,  $T_s$  is surface temperature and  $\theta$  is the zenith angle. (**Revised manuscript lines 97-100**)

- 6. Line 161, ERA5 has 37 fixed pressure levels vertically, and 137 model levels distributed using hybrid sigma-pressure coordinate system. It seems like you're using the model levels. It is recommended to point this out explicitly in the manuscript.
- Response: Thank you for your suggestions. We utilized ERA5 hourly specific humidity data on pressure levels from 1940 to present at the 400 hPa level from

the 37 fixed pressure levels of ERA5. This data was used to validate that our selected channels combination effectively removes the influence of water vapor interference in sulfur dioxide monitoring. The ERA5 data used in this study can be accessed via the Copernicus Climate Change Service (C3S) Climate Data Store (CDS; https://doi.org/10.24381/cds.adbb2d47). In this revision, we have provided a detailed description of the data types utilized as follows to give a clearer understanding:

Each profile from ERA5 has a horizontal scale of 31 km. and This includes upper-air parameters on 37 fixed pressure levels from 1,000 to 1 hPa and 137 model levels distributed using hybrid sigma-pressure coordinate system.137 vertical levels, ranging from near-surface air pressure to 0.01 hPa. For this study, we interpolate ERA5 400 hPa fixed pressure level data to assess atmospheric water vapor conditions near Mount Ruang concurrent with FY-3E/HIRAS-II observations. (**Revised manuscript line 167**)

- 7. Figure 4, on the figure it seems like the selection of water vapor channels only depends on cross-comparison with temperature channels. According to lines 219 to 220, with the selected SO2 channels being a subset that aligns with the water vapor channels (purple links), there should also be a cyan link between water vapor channels and SO2 channels which points to water vapor selections. Or as illustrated in the figure, the relevant contents should be like 'the water vapor Jacobian of SO2 channels must match those of the water vapor channel, while the temperature Jacobian of water vapor channels must match those of the SO2 channels.'
- Response: Thank you for your feedback. We fully concur with the reviewer's observations and have comprehensively revised Figure 4 to address these comments. In the updated figure:
  - 1. SO2 channel selection was guided by Jacobian analysis.

2. Atmospheric temperature channels were determined through comparative analysis of temperature Jacobians between the atmospheric temperature absorption region and the pre-selected SO2 channels.

3. Water vapor channel selection employed a two-stage process:

First, temperature Jacobians from the water vapor absorption region were cross-compared with those from SO2 channels

Second, water vapor Jacobians from the same region were analyzed against corresponding Jacobians from SO2 channels.

This systematic approach yielded optimal water vapor channel selections. Note that the selected water vapor channels with similar temperature and water vapor Jacobians of SO2 channels do not contain SO2 absorption, meaning there is no overlapping channel between selected water vapor channels and SO2 channels. (Revised manuscript Section 3.2, Figure 4).

Figure 4: Schematic diagram of channel selection method.

- Line 232, it seems like the additional SO2 signal is around 1125 cm-1 rather than 1225 cm-1 from Figure 5.
- Response: Thank you for pointing out the problem. To more comprehensively characterize the absorption capacity of satellite channels for SO2, we have expanded the SO2 absorption range in our study to 1100 1430 cm-1. Additionally, although the SO2 signal in the 1100 1170 cm-1 range is relatively weak, it remains significant for monitoring tropospheric SO2. Therefore, we have also included SO2 monitoring channels within the 1100 1170 cm-1 range to achieve more accurate SO2 detection. (Revised manuscript Sec. 3.2.1, Figure 5)

Figure 5: Schematic diagram of the SO2 Jacobian matrix with atmospheric profiles from the US Standard Atmosphere, 1976.

- 9. There should be another set of figures between Figure 6 and 7 showing the temperature Jacobian functions of the channels within SO2 absorption region.
- **Response**: Thank you very much for the constructive feedback. We have redefined the spectral ranges for the SO2 absorption region and the water vapor absorption region. Following your suggestion, we have supplemented the sections on atmospheric temperature channel selection and water vapor channel selection with Jacobian figures for atmospheric temperature and water vapor within the SO2 absorption region, respectively. In these figures, we have annotated the SO2 channels selected in the study to clearly indicate their Jacobian peak values. The following are the temperature Jacobian figures and water vapor Jacobian figures for the SO2 absorption region. (**Revised manuscript Sec. 3.2.2 and Sec. 3.2.3**)